# LEARNING-AUGMENTED LEARNING OF GAUSSIAN MIXTURE MODELS

## ABSTRACT

Gaussian mixture models (GMMs) is one of the most fundamental methods to identify and extract latent structure in complex datasets. Unfortunately, well-known hardness results require that any algorithm for learning a mixture of $k$ multivariate Gaussian distributions in $d$-dimensional space requires both runtime and sample complexity exponential in $d$, even if the Gaussians are reasonably separated. To overcome this barrier, we consider settings where algorithms are augmented with possibly erroneous "advice" to help learn the underlying GMMs. In particular, we consider a natural predictor that can be easily trained through machine learning models. Specifically, our predictor outputs a list of $\beta$ possible labels for each sample from the mixture such that, with probability at least $1 - \alpha$, one of the labels in the list is the true label, for a fixed constant $\alpha$. We show that to estimate the mixture up to total variation distance $\tilde{\mathcal{O}}(\varepsilon)$, we can use $k \cdot \text{poly}\left(d, \log k, \frac{1}{\varepsilon}\right)$ samples from the GMM, provided that $\beta$ is upper bounded by any fixed constant. Moreover, our algorithm uses polynomial time, thus breaking known computational limitations of algorithms that do not have access to such advice.

## 1 INTRODUCTION

The problem of learning a model that best fits a collection of observations is a fundamental problem in machine learning, theoretical computer science, and mathematical statistics. A common assumption is that the input data is generated through independent and identically distributed samples from an unknown mixture of Gaussians. Formally, a Gaussian mixture model (GMM) is a convex combination of Gaussian distributions, i.e., a distribution $\mathcal{D} = \sum_{i=1}^{k} w_i \cdot \mathcal{N}(\mu_i, \Sigma_i)$, where the unknown weights $w_i$ are non-negative and sum to 1 and $\mathcal{N}(\mu_i, \Sigma_i)$ denotes a $d$-dimensional multivariate Gaussian distribution with unknown mean $\mu_i$ and covariance matrix $\Sigma_i$. Due to their capacity to model diverse data distributions with a combination of Gaussian components, GMMs have been one of the most extensively studied latent variable model since its introduction by Pearson in 1894 (Pearson, 1894), with applications in many areas where finite mixture models occur, including biology, geology, physics, and social sciences (Titterington et al., 1985; Peel & MacLahlan, 2000; McLachlan et al., 2019).

Dasgupta (1999) initiated the algorithmic problem of learning GMMs by giving an algorithm for learning the mixture of $k$ identical Gaussians that uses time polynomial in $k$ and the ambient dimension $d$, under the assumption that the Gaussians are spherical and their centers are well-separated. Subsequent works (Arora & Kannan, 2001; Vempala & Wang, 2002; Achlioptas & McSherry, 2005; Brubaker & Vempala, 2008; Kannan et al., 2008; Diakonikolas et al., 2023) focused on efficient algorithms for learning GMMs under various other assumptions. Additionally, a line of work (Kalai et al., 2010; Moitra & Valiant, 2010; Belkin & Sinha, 2015; Hardt & Price, 2015) studied efficient algorithms and impossibility results under minimal information-theoretic conditions.

Unfortunately, Moitra & Valiant (2010) showed that there exist distributions $\mathcal{D}_1$ and $\mathcal{D}_2$ that are mixtures of $k^2 + 1$ Gaussians with "large" weights and reasonably separated centers, such that $2^{\Omega(k)}$ samples are needed to distinguish between $\mathcal{D}_1$ and $\mathcal{D}_2$. Moreover, Hardt & Price (2015) showed that even for the case where $d = 1$ and the Gaussians have the same variance, $\Omega(\sigma^{6k-2})$ samples are necessary to learn the parameters in the mixture model up to constant additive error, where $\sigma^2$ is the variance of the univariate Gaussians. These results say that both sample complexity and runtime

exponential in $k$ is necessary for learning general Gaussian mixture models. Thus, we seek tools and techniques that enable sample complexity and runtime polynomial in $k$ for learning GMMs without require distributional assumptions beyond the necessary minimal information theoretic assumptions.

**Learning-augmented algorithms.** A natural area to draw inspiration for new tools is the recent advances in the predictive ability of machine learning models. In many applications, auxiliary information, e.g., previous datasets with potentially similar behavior, is often available and can guide algorithmic decisions if accurate. On the other hand, machine learning models lack provable guarantees and thus produce heuristics that can be embarrassingly inaccurate when generalizing to unfamiliar inputs (Szegedy et al., 2014). Nevertheless, *learning-augmented algorithms* (Mitzenmacher & Vassilvitskii, 2020) have been shown to overcome worst-case impossibility barriers for a wide range of settings, such as more efficient data structures (Kraska et al., 2018; Mitzenmacher, 2018; Lin et al., 2022), faster runtime algorithms (Dinitz et al., 2021; Chen et al., 2022c; Davies et al., 2023), more competitive online algorithms (Purohit et al., 2018; Gollapudi & Panigrahi, 2019; Lattanzi et al., 2020; Wang et al., 2020; Wei & Zhang, 2020; Bamas et al., 2020; Im et al., 2021; Lykouris & Vassilvitskii, 2021; Aamand et al., 2022; Anand et al., 2022; Azar et al., 2022; Grigorescu et al., 2022; Khodak et al., 2022; Jiang et al., 2022; Antoniadis et al., 2023; Shin et al., 2023), and more space-efficient streaming algorithms (Hsu et al., 2019; Indyk et al., 2019; Jiang et al., 2020; Chen et al., 2022b;a; Li et al., 2023). In particular, (Ergun et al., 2022; Nguyen et al., 2023) introduce algorithms for $k$-means and $k$-median clustering that use polynomial runtime and achieve approximation guarantees beyond NP hardness limits. As clustering has many similar structural properties as, and in fact is often used as a subroutine for, learning GMMs, it is our hope that we can also use machine learning advice to overcome the exponential time barriers for learning GMMs (Moitra & Valiant, 2010; Hardt & Price, 2015).

## 1.1 OUR CONTRIBUTIONS

In this paper, we study the problem of learning Gaussian mixture models given a natural form of advice that can be readily offered by machine learning models.

**List oracle.** We consider learning a mixture $\mathcal{D}$ of $k$ Gaussians given a *list oracle*, which provides a list of labels for each samples, i.e., a list of possible indices of the corresponding Gaussian, up to some error $\alpha$. Specifically, if $x \sim \mathcal{D}$ is drawn from Gaussian $G_i = (\mu_i, \Sigma_i)$, then informally, on query $x$, the oracle will provide a small list of labels that includes the label $i$, with probability at least $(1 - \alpha)$ over the fraction of the input queries. In particular, we only require a certain fraction of the labels to be incorrect, regardless of the manner they are incorrect. That is, the labels can be adversarially incorrect, which could potentially obfuscate any signal from a mixture of Gaussians that are not well-separated.

We remark that a list oracle can be easily acquired through a machine learning model that is trained on a similar distribution of data. For example, a set of samples can be split into a "training" set and a "testing" set, but rather than using the training set in the traditional line of supervised learning, we can instead apply a heuristic such as `kmeans++` to cluster the initial data, and use the resulting centers to form a predictor for the second half of the data. The model may assign different confidences to each label for a point, e.g., an algorithm may declare $55\%$ confidence that an input point $x$ is labeled $i$ and $45\%$ confidence that $x$ should be labeled $j$. In this case, it seems reasonable to assign both $i$ and $j$ as possible labels for $x$. Alternatively, consider the scenario where we have multiple machine learning models that can be used to predict the label of a sample. We could use ensemble learning to assimilate the multiple models into a single model, but this could lose valuable information, e.g., if half of the models declare that an input point $x$ is labeled $i$ and half of the models declare that $X$ is labeled $j$. In fact, note that it is possible for an ensemble of 10 (and more generally $\beta$) models to always correctly predict a label despite each of models only have $10\%$ (and more generally $\frac{1}{\beta}$) accuracy.

We also remark that a list oracle does not trivialize the problem. For a simple example, consider a mixture of $k = 2$ Gaussians and a list oracle that simply outputs both labels for each point. In this case, the list oracle provides absolutely no additional information at all! More generally, the list oracle could produce lists so that all the points in Gaussians $i$ and $j$ are given the same labels, in which case any signal separating $i$ and $j$ seems to be lost.

Furthermore, even if the list oracle only contains a *single* label that is correct with arbitrarily high accuracy, the probelm is surprisingly not immediate, in the sense that the naïve algorithm outputting the empirical mean-covariance pairs of the labeled samples does not work. In fact, the algorithm can perform arbitrarily poorly. As a simple example, consider the case where $k = 2$, so the goal is to learn a mixture of two Gaussians $G_1$ and $G_2$ using $n$ samples, given an oracle with accuracy $1 - \frac{3}{n}$, so that the error rate $\alpha = \frac{1}{n}$ is arbitrarily low. Suppose $G_1$ has uniform spherical covariance and is centered at the origin, while $G_2$ has uniform spherical covariance and is centered at the $N \cdot \mathbf{e}_1$, for the elementary vector $\mathbf{e}_1$. With constant probability, the oracle labels all points correctly except for a single point. Suppose without loss of generality, the oracle mistakenly labels a sample $x$ from $G_1$ as having been generated from $G_2$. Since $G_1$ is centered at the origin, then $x$ has distance $\mathcal{O}(N)$ from $N \cdot \mathbf{e}_1$, and thus the empirical mean of the cluster of points labeled as being generated from $G_2$ has changed additively by $\mathcal{O}\left(\frac{N}{n}\right)$. For $N \gg n$, the empirical mean of the cluster can be arbitrarily far from the true mean of $G_2$, despite only a single error by the oracle. Thus even though the example can be quite easily rectified, it is apparent that blindly outputting the empirical means and covariances of the induced clustering by the list oracle can give arbitrarily large error, even when the error is arbitrarily small. We observe that the above example of course does not rule out more complex algorithms.

We first consider the case where the list oracle always includes the correct label, i.e., $\alpha = 0$; we subsequently discuss the case for general $\alpha$. We show that a list oracle with $\beta$ labels, in addition to $k \cdot \mathrm{poly}\left(d, \log k, \frac{1}{\varepsilon}\right)$ samples from the mixture $\mathcal{D}$ of $k$ well-separated Gaussians, can be used to output a mixture $\mathcal{D}'$ of $k$ Gaussians such that $d_{\mathsf{TV}}(\mathcal{D}, \mathcal{D}') \leq \tilde{\mathcal{O}}(\varepsilon)$. Formally, we require bounds on the precision and recall of the set of labels for each cluster. This is to avoid cases, such as where two clusters $i$ and $j$ are completely obfuscated because all lists that include the label $i$ also include the label $j$, and vice versa. We defer the formal description of precision and recall to Definition 2.4. For the purposes of presentation, we represent our result for uniform mixtures, though we note that our result extends to non-uniform mixtures, with the appropriate increase in runtime, sample complexity, and changes in the total variation distance between pairs $G_i$ and $G_j$ of Gaussians in the mixture model.

**Theorem 1.1.** *Let $\mathcal{D}$ be any $d$-dimensional uniform mixture of Gaussians $G_i, \ldots, G_k$ with $-\log(1 - d_{\mathsf{TV}}(G_i, G_j)) = \Omega(\log(k/\varepsilon))$ for all $i \neq j$. Given a $\beta$-list oracle, there exists an algorithm that takes $n = \mathrm{poly}(dk/\varepsilon)$ samples from $\mathcal{D}$, runs in time $k \cdot \mathrm{poly}(n)$, and returns $k$ hypothesis Gaussians $H_1, \ldots, H_k$ such that with high probability,*

$$d_{\mathsf{TV}}\left(\frac{1}{k}\sum_{i \in [k]} G_i, \frac{1}{k}\sum_{i \in [k]} H_i\right) \leq \tilde{\mathcal{O}}(\varepsilon).$$

The algorithm for Theorem 1.1 first looks at the sets $P_i$ of points labeled $i$, for each $i \in [k]$. Under bounded recall assumptions, we have that the number of false positives in $P_i$ is a large constant multiple of the number of points that were truly sampled from $G_i$. We thus apply a list-decodable mean and covariance estimation routine to acquire a list $\mathcal{H}_i$ of mean-covariance pairs, i.e., Gaussians, including a pair that is close in total variation distance to the true Gaussian $G_i$.

Unfortunately, $\mathcal{H}_i$ may also include a number of incorrect mean-covariance pairs, or even mean-covariance pairs corresponding to other Gaussians $G_j$. A natural approach would be to merge the similar pairs and then remove the remaining false mean-covariance pairs through a standard tournament, e.g., Daskalakis et al. (2015); De et al. (2015). However, because each index $i \in [k]$ corresponds to a list with a constant size, the number of possible hypotheses is $2^{\Omega(k)}$, which would require an additional number of samples exponential in $k$, to determine the correct hypothesis.

We instead recall the insight by Bakshi et al. (2020) that a false mean-covariance pair $(\mu, \Sigma)$ must be separated from a true Gaussian $G_i$ in *parameter distance*, i.e., large total variation implies large parameter distance. In particular, $(\mu, \Sigma)$ must be mean-separated, Frobenius-separated, or spectrally-separated; surprisingly, there are no other cases. See Figure 1 for an illustration.

We show that under each of the above cases, we can use a maximum likelihood estimator to assign each sample, resulting in the true mean-covariance pairs having a large number of samples and the false mean-covariance pairs having a small number of samples. Specifically, we prove a structural property showing that large parameter distance for Gaussians implies large total variation distance,

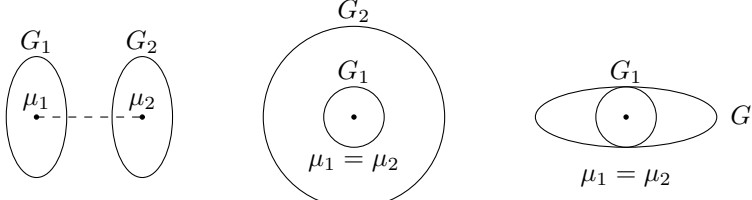

Fig. 1: From left to right: examples of mean-separation, Frobenius-separation, and spectral-separation.

i.e., the converse of the statement of Bakshi et al. (2020). To that end, we consider casework on whether a pair of Gaussians $G_i$ and $G_j$ are mean-separated, Frobenius-separated, or spectrally-separated. If the Gaussians are mean-separated, we note there exists a vector $v$ that has significant correlation with the difference $\mu_i - \mu_j$ between the two means $\mu_i, \mu_j$ of the Gaussians, so that a sample $x$ can be distinguished as being drawn from $G_i$ or $G_j$ depending on the values of $\langle x - \mu_i, v \rangle$ and $\langle x - \mu_j, v \rangle$. If the Gaussians are Frobenius-separated, we show that the eigenvalues of the matrix $\Sigma_j^{-1/2} \Sigma_i \Sigma_j^{-1/2}$ can be used to show that Gaussians are separated in total variation distance. Finally, if the Gaussians are spectrally-separated, we prove that there exists a direction $v$ such that the projection of $x - \mu_i$ and $x - \mu_j$ onto $v$ can distinguish whether $x$ was drawn from $G_i$ or $G_j$ with high probability. We remark that large parameter distance between two distributions does not imply large total variation distance between the distribution sin general, e.g., Bakshi et al. (2020) give two sub-Guassian distributions that have total variation distance $1 - \Omega(1)$ but arbitrarily large parameter distance. Therefore, our proof crucially lies on properties of Gaussians that are absent from general distributions.

**List oracle with errors.** We next address the case where the true label may not be among the $\beta$ labels output by the list oracle for each sampled point. We remark that our techniques can be easily extended to the case where the correctness of the oracle only holds to some sufficiently small but constant error rate $\alpha$. More specifically, we simply require that for $1 - \alpha$ fraction of the points, the correct label is among the $\beta$ predicted labels for each sample, while for the remaining $\alpha$ fraction of the points, the $\beta$ predicted labels does not need to include the correct label.

**Theorem 1.2.** *Let $\mathcal{D}$ be any $d$-dimensional uniform mixture of Gaussians $G_1, \ldots, G_k$ with $-\log(1 - d_{\mathsf{TV}}(G_i, G_j)) = \Omega(\log(k/\varepsilon))$ for all $i \neq j$. Given a $\beta$-list oracle with a sufficiently small constant error rate, there exists an algorithm that takes $n = \operatorname{poly}(dk/\varepsilon)$ samples from $\mathcal{D}$, runs in time $k \cdot \operatorname{poly}(n)$, and returns $k$ hypothesis Gaussians $H_1, \ldots, H_k$ such that with high probability,*

$$d_{\mathsf{TV}}\left(\frac{1}{k}\sum_{i \in [k]} G_i, \frac{1}{k}\sum_{i \in [k]} H_i\right) \leq \tilde{\mathcal{O}}(\varepsilon).$$

Toward Theorem 1.2, the main observation is that the list decoding algorithm is robust to an $\alpha \in (0, 1)$ fraction of corruptions when $\alpha$ is at most a fixed small constant. We provide additional details in Section 3. Furthermore, we also note that in the case $\beta = 1$, so that each query is only given a single label, there exists a significantly simpler learning-augmented algorithm using the techniques of Diakonikolas et al. (2020) to prune away notable outliers. We describe this simpler approach in Appendix B.

**Robustness.** Thus far, our discussion has centered around algorithmic guarantees that are smooth with increasingly accurate oracle predictions. However, it also may be the case where the oracle predictions are completely nonsensical. We remark that because we are studying an offline setting, robustness follows immediately from running a classical mixture learning algorithm in addition to our learning-augmented algorithm and choosing the better of the two outputs. In particular, if the output of the classical mixture learning algorithm is $\mathcal{D}_1$ and the output of the learning-augmented algorithm is $\mathcal{D}_2$, and these two mixtures have large total variation distance, then it is possible to generate an additional number of samples and run a standard tournament procedure, e.g., (Daskalakis et al., 2015; De et al., 2015), to choose the better fit of the two mixtures.

**Empirical evaluations.** We complement our theoretical results with a number of experimental results on learning Gaussian mixture models, comparing the performance of a scaled-down version of our learning-augmented algorithm with the standard clustering baseline Lloyd's algorithm. We generate a number of synthetic datasets from mixture distributions $\mathcal{D}$ that are a uniform mixture of $k$ Gaussians that are not well-separated, as otherwise existing techniques can already efficiently learn the Gaussians. We compare the number of points that are correctly labeled by both the baseline algorithm and our learning-augmented algorithm, as well as plot the resulting cluster induced by both algorithms. Our results demonstrate that our learning-augmented algorithm can both produce more accurate classification than the standard baseline across various parameter settings, as well as produce clusterings that are more aligned with mixtures of Gaussians. Overall, our empirical evaluations indicate that our learning-augmented algorithm performs effectively in practice, further supporting its theoretical guarantees. We present these results in Section 4.

## 2 MODEL

In this section, we formally introduce the list model oracle; the reader may wish to consult Appendix A for additional notation and preliminaries. For a vector $\mu \in \mathbb{R}^d$ and a positive-definite matrix $\Sigma \in \mathbb{R}^{d \times d}$, we use $\mathcal{N}(\mu, \Sigma)$ to denote a Gaussian with mean $\mu$ and covariance $\Sigma$, which has the following probability density function.

**Definition 2.1** (Multivariate Gaussian). *For $x \in \mathbb{R}^d$, the probability density function $p(x)$ of $\mathcal{N}(\mu, \Sigma)$ is*

$$p(x) = (2\pi)^{-k/2} \det(\Sigma)^{-1/2} \exp\left(-\frac{1}{2}(x - \mu)^\top \Sigma^{-1}(x - \mu)\right).$$

We recall the following fact about the distribution of the projection of a multivariate Gaussian onto a unit vector.

**Fact 2.2.** *Given a sample $x$ from Gaussian $\mathcal{N}(\mu, \Sigma)$ and a unit vector $v \in \mathbb{R}^d$, we have that $\mathrm{Proj}(x, v)$ is distributed as $\mathcal{N}(v^\top \mu, v^\top \Sigma v)$.*

*Proof.* Let $x \sim \mathcal{N}(\mu, \Sigma)$. Then the projection of $x$ onto the subspace given by $v$ is $v(v^\top v)^{-1}v^\top x = (v^\top x)v$, so that $v^\top x$ is a scalar and $v^\top x v$ is the corresponding vector in the direction of $v$. Note that this is a linear transformation of a Gaussian, so that $v^\top x \sim \mathcal{N}(v^\top \mu, v^\top \Sigma v)$. □

We recall the following impossibility result for learning a mixture of $k$-Gaussians by Moitra & Valiant (2010).

**Theorem 2.3.** *(Moitra & Valiant, 2010) There exist distributions $\mathcal{D}_1$ and $\mathcal{D}_2$ on $\mathbb{R}$ that are mixtures of $k^2 + 1$ Gaussians, such that*

$$\|\mathcal{D}_1 - \mathcal{D}_2\|_1 \leq 11ke^{-k^2/24}.$$

*Moreover, the weights of each mixture are at least $\frac{1}{4k^2+2}$ and the centers for each Gaussian are separated by at least $\frac{1}{k}$.*

Theorem 2.3 implies that $2^{\Omega(k)}$ samples are needed to distinguish between $\mathcal{D}_1$ and $\mathcal{D}_2$, even when the weights of the mixture are relatively large and the centers of the Gaussians are well-separated. Similarly, Theorem 2.3 shows that at least $2^{\Omega(k)}$ runtime is necessary, which is exponential in $k$. In this paper, our goal is to design algorithms that bypass these exponential time and sample complexity hardness results with the aid of additional possibly erroneous advice.

We consider the following formulation of a list oracle, which outputs a list of possible labels for each query point.

**Definition 2.4** (List oracle). *Given a set $X$ of $n$ samples $x_1, \ldots, x_n \subset \mathbb{R}^d$ from a mixture $\mathcal{D}$ of $k$ Gaussians $G_1, \ldots, G_k$. For each point $x_i$, a list oracle with $\beta$ labels outputs a list of $\beta$ labels, which includes the label $j \in [k]$ for which $x_i$, where $\beta \geq 1$ is a list-decoding rate for the predictor.*

A list oracle can also simply be generated by a machine learning model by asking the model to output the $\beta$ labels for which it has the largest confidence, rather than just the singular label for which it has the highest confidence. More generally, we say a list oracle has error rate $\alpha$ if the list only contains the true label across $1 - \alpha$ fraction of all queries.

## 3 LIST ORACLES

In this section, we give an algorithm that uses a list oracle and learns a mixture of $k$ Gaussians up to arbitrary total variation distance $\tilde{\mathcal{O}}(\varepsilon)$ given a polynomial number of additional samples and runtime. We first require the following definition of parameter distance, which allows us to separate pairs of Gaussians as being either mean-separated, covariance/Frobenius-separated, or spectrally-separated, e.g., see Figure 1.

**Definition 3.1** (Parameter Distance). *Given $\Delta \geq 1$, we say that the parameter distance between mean-covariance pairs $(\mu_1, \Sigma_1)$ and $(\mu_2, \Sigma_2)$ is at most $\Delta$, i.e, $d_{\mathsf{Param}}((\mu_1, \Sigma_1), (\mu_2, \Sigma_2)) \leq \Delta$ if:*

- *(Mahalanobis Mean Closeness) For all $v \in \mathbb{R}^d$, we have $\langle \mu_1 - \mu_2, v \rangle^2 \leq \Delta v^\top (\Sigma_1 + \Sigma_2) v$*

- *(Multiplicative Spectral Closeness) For all $v \in \mathbb{R}^d$, we have $\frac{1}{\Delta} v^\top \Sigma_2 v \leq v^\top \Sigma_1 v \leq \Delta v^\top \Sigma_2 v$*

- *(Relative-Frobenius Closeness) $\left\| \Sigma_1^{\dagger/2} \Sigma_2 \Sigma_1^{\dagger/2} - I \right\|_F \leq \Delta$.*

We recall the following statement that given any two Gaussian distributions with parameter distance upper bounded by $\Delta$, then their total variation distance must also be upper bounded by a fixed function of $\Delta$.

**Lemma 3.2** (Proposition A.1 in Bakshi et al. (2020), Fact 3.24 in Ivkov & Kothari (2022)). *For $\Delta \geq 1$, suppose $\mu_1, \mu_2$ and $\Sigma_1, \Sigma_2$ satisfy $d_{\mathsf{Param}}((\mu_1, \Sigma_1), (\mu_2, \Sigma_2)) \leq \Delta$. Then $d_{\mathsf{TV}}(\mathcal{N}(\mu_1, \Sigma_1), \mathcal{N}(\mu_2, \Sigma_2)) \leq 1 - \exp(-\mathcal{O}(\Delta^2 \log \Delta))$.*

We prove that the converse must also hold, in the sense that two Gaussian distributions with parameter distance lower bounded by $\Delta$ must also have their total variation distance lower bounded by a fixed function of $\Delta$.

**Lemma 3.3.** *For $\Delta \geq 1$, suppose $\mu_1, \mu_2$ and $\Sigma_1, \Sigma_2$ satisfy $d_{\mathsf{Param}}((\mu_1, \Sigma_1), (\mu_2, \Sigma_2)) > \Delta$. Then $d_{\mathsf{TV}}(\mathcal{N}(\mu_1, \Sigma_1), \mathcal{N}(\mu_2, \Sigma_2)) \geq 1 - \frac{1}{\mathrm{poly}(\Delta)}$.*

Lemma 3.3 is perhaps most technical contribution of our paper; we prove Lemma 3.3 in Appendix C.1 by separately considering Gaussians pairs $(\mu_1, \Sigma_1)$ and $(\mu_2, \Sigma_2)$ that are mean-separated, Frobenius-separated, and spectrally-separated. Specifically, we show in Lemma C.1 that if $(\mu_1, \Sigma_1)$ and $(\mu_2, \Sigma_2)$ are mean-separated, then there exists an explicit vector $v$ that can be used to classify the sample $x$, based on $\langle x - \mu_1, v \rangle$ and $\langle x - \mu_2, v \rangle$. We show in Appendix C.1.2 that if $(\mu_1, \Sigma_1)$ and $(\mu_2, \Sigma_2)$ are Frobenius-separated, then the eigenvalues of the matrix $\Sigma_j^{-1/2} \Sigma_i \Sigma_j^{-1/2}$ can be used lower bound the total variation distance. Finally, we show in Lemma C.7 that if the Gaussians are spectrally-separated, then there exists a direction $v$ such that the projection of $x - \mu_1$ and $x - \mu_2$ onto $v$ can distinguish whether $x$ was drawn from $(\mu_1, \Sigma_1)$ and $(\mu_2, \Sigma_2)$ with high probability.

To ensure that our Gaussians are well-separated in parameter distance, we first utilize the following list-decoding algorithm that returns a list of candidate centers.

**Theorem 3.4** (Theorem 1.4 in Ivkov & Kothari (2022)). *Given ambient dimension $d$ and corruption rate $\alpha$, let $n = d^{\alpha^{-\mathcal{O}(1)}}$. Then there exists an algorithm that takes $n$ samples with corruption rate $(1 - \alpha)$ from a dimension $d$ Gaussian distribution with mean $\mu$ and covariance $\Sigma$ and uses $n^{\mathrm{poly}(1/\alpha)}$ time to output a list $\mathcal{H}$ of $H(1/\alpha) := 2^{\mathcal{O}(1/\alpha^{\mathcal{O}(1)})}$ Gaussians such that with probability at least 0.99 (over the drawn samples and the randomness of the algorithm), there exists $(\widehat{\mu}, \widehat{\Sigma}) \in \mathcal{H}$ with $d_{\mathsf{Param}}((\mu, \Sigma), (\widehat{\mu}, \widehat{\Sigma})) \leq Z_1(1/\alpha)$, for some fixed polynomial $Z_1$, i.e., $d_{\mathsf{TV}}(\mathcal{N}(\mu, \Sigma), \mathcal{N}(\widehat{\mu}, \widehat{\Sigma})) \leq 1 - \exp(-\alpha^{-\mathcal{O}(1)})$.*

Given Theorem 3.4, the natural approach would to be procure a large number of samples and then perform list-decoding on the set of points for each label. However, this could result in a number of redundancies. For example, consider the case where almost all the points from $(\mu_1, \Sigma_1)$ and $(\mu_2, \Sigma_2)$ are both given labels 1 and 2. Then roughly the same Gaussian will be returned by the list decoding procedure on the points labeled 1 and the points labeled 2. Hence, we require the subroutine in Algorithm 1 to merge close pairs of Gaussians.

We first observe that the Gaussians in the list $\mathcal{L}$ output by Algorithm 1 are pairwise well-separated in parameter distance.

---

**Algorithm 1** Algorithm MERGECLOSEPAIRS($\mathcal{L}$) to merge close pairs in a list of Gaussians

---

**Input:** List $\mathcal{L}$ of Gaussians $\{(\mu_i, \Sigma_i)\}$, parameter $\Delta > 0$
**Output:** Pruned list $\mathcal{L}$ with well-separated Gaussians
1: **for** $i \in [|\mathcal{L}|]$ **do**
2:     Initialize group $\mathcal{G}_i = (\mu_i, \Sigma_i)$
3: **end for**
4: **for** $i \in [|\mathcal{L}|]$ **do**
5:     **for** all $a \neq b$ with $(\mu_i, \Sigma_i) \in \mathcal{G}_a$, $(\mu_j, \Sigma_j) \in \mathcal{G}_b$ and $d_{\mathsf{Param}}((\mu_i, \Sigma_i), (\mu_j, \Sigma_j)) \leq \Delta$ **do**
6:        Merge groups $\mathcal{G}_a$ and $\mathcal{G}_b$
7:     **end for**
8: **end for**
9: Return $\mathcal{L}$

---

**Observation 3.5.** *[Well-separatedness of $\mathcal{L}$] Let $\mathcal{L} = \{(\mu_i, \Sigma_i)\}$ be the output of* MERGECLOSEPAIRS. *Then for each $i \neq j$, we have $d_{\mathsf{Param}}((\mu_i, \Sigma_i), (\mu_j, \Sigma_j)) > \Delta$.*

Given Observation 3.5, we can now form a list of well-separated candidate Gaussians. However, some of these may be false positives induced by erroneous labels. On the other hand, by standard concentration inequalities, we have rough lower bounds on the true number of points sampled from each Gaussian. Therefore, if we could approximately match each sample to the candidate Gaussian that most likely generated the sample, then the resulting false candidate Gaussians would not be matched with enough samples. Hence, we utilize the following subroutine in Algorithm 2 to partially cluster the sampled points.

---

**Algorithm 2** Algorithm PARTIALCLUSTER to partition points into well-separated clusters

---

**Input:** Input set $X$, list $\mathcal{L}$ of weighted Gaussians $\{w_i, \mathcal{N}(\mu_i, \Sigma_i)\}$
**Output:** Clustering $\{P_i\}$ of subset of $X$, i.e., partial clustering of $X$
1: $\mathcal{K} \leftarrow |\mathcal{L}|$
2: $\mathcal{P} \leftarrow \emptyset$, $\mathcal{P}_i \leftarrow \emptyset$ for all $i \in [\mathcal{K}]$
3: Let $p_i(x)$ be the log-likelihood function of $x$ for $\mathcal{N}(\mu_i, \Sigma_i)$
4: **for** $x \in X$ **do**
5:     Let $j = \operatorname{argmax}_{i \in [\mathcal{K}]} p_i(x)$          ▷Break ties arbitrarily
6:     $\mathcal{P}_j \leftarrow \mathcal{P}_j \cup \{x\}$
7: **end for**
8: Return $\mathcal{P}$

---

We observe that each point in the initial sample $X$ is given at most one label among $[\mathcal{K}]$ by Algorithm 2.

**Observation 3.6.** *[Partial clustering of $X$] Let $x \in X$ and let $\mathcal{P} = \{P_1, \ldots, P_{|\mathcal{P}|}\}$ be the output of* PARTIALCLUSTER$(X, \mathcal{L})$ *for any $\mathcal{L}$. Then there exists exactly one index $i \in [|\mathcal{P}|]$ such that $x \in P_i$.*

We now give our main algorithm that applies our earlier strategy. We first acquire a large number of samples and form a list of candidate Gaussians by performing list-decoding on the set of sampled points with each label. We then partially cluster the samples, removing the candidate Gaussians without a sufficiently high number of points. The algorithm appears in full in Algorithm 3.

To show correctness of Algorithm 3, we require both soundness and completeness. That is, we must first show that all false candidate Gaussians do not receive enough samples after the partial clustering procedure. We then must also show that for each underlying Gaussian, there is a candidate Gaussian generated by the list-decoding procedure that is close in parameter distance and also receives a sufficient number of samples after the partial clustering procedure. For these properties, we crucially use the parameter distance characterization of Gaussian mixtures.

We first require the following property showing "near-transitivity" of Gaussians that are close under parameter distance.

---

**Algorithm 3** Learning-augmented algorithm for learning mixture of $k$-Gaussians, multiple labels

---

**Input:** List-decoding rate $\beta > 0$, set $X$ of $k \cdot \text{poly}(d/\varepsilon)$ samples $s_1, \ldots, s_n \subset \mathbb{R}^d$ from a mixture
    $\mathcal{D}$ of $k$ Gaussians $G_1, \ldots, G_k$
**Output:** $\widetilde{\mathcal{D}}$ with $d_{\mathsf{TV}}(\mathcal{D}, \widetilde{\mathcal{D}}) \leq \tilde{\mathcal{O}}(\varepsilon)$
1: Let $\zeta > 1$ be a sufficiently large constant          ▷See Lemma 3.7
2: $\gamma \leftarrow \frac{1}{2}$
3: $\mathcal{L} \leftarrow \emptyset, \xi \leftarrow H(1/\beta)$ and $\Delta \leftarrow Z_1\left(\frac{1}{\xi}\right)\left(\frac{\xi k}{\gamma}\right)^{\zeta}$, for the functions $H$ and $Z_1$ from Theorem 3.4
4: **for** $i \in [k]$ **do**
5:     Let $\widetilde{P}_i$ be the set of points that have label $i$
6:     Let $\mathcal{H}_i = \{(\mu_j, \Sigma_j)\}_j$ be the output of the list-decoding algorithm on $\widetilde{P}_i$ with parameter $\frac{1}{\beta}$
7:     $\mathcal{L} \leftarrow \mathcal{L} \cup \mathcal{H}_i$
8: **end for**
9: $\mathcal{L} \leftarrow \text{MERGECLOSEPAIRS}(\mathcal{L})$
10: Partition $\mathcal{L}$ into groups $\mathcal{G}_1, \mathcal{G}_2, \ldots$ such that for all groups $\mathcal{G}_a$ and $\mathcal{G}_b$, there does not exist
     $(\mu_1, \Sigma_1) \in \mathcal{G}_a$ and $(\mu_2, \Sigma_2) \in \mathcal{G}_b$ with $a \neq b$ such that $d_{\mathsf{Param}}((\mu_1, \Sigma_1), (\mu_2, \Sigma_2)) \leq \Delta$
11: $\widetilde{\mathcal{P}} \leftarrow \text{PARTIALCLUSTER}(X, \mathcal{L})$
12: Delete from $\widetilde{\mathcal{P}}$ the groups of $\mathcal{L}$ with less than $\frac{1}{2k}$ fraction of the samples in $X$
13: Let $\widetilde{\mathcal{D}} = \{H_i\}_{i \in [k]}$ be the output of ROBUSTGAUSSIANS on $\widetilde{\mathcal{P}}$ with accuracy $\varepsilon$ ▷Theorem B.5
14: Output $\widetilde{\mathcal{D}}$

---

**Lemma 3.7** (Claim 6.3 in Liu & Moitra (2021)). *Let $G_1, G_2, G_3$ be Gaussians such that $G_1$ and $G_2$ are $C$-close and $G_2$ and $G_3$ are $C$-close in parameter distance. Then $G_1$ and $G_3$ are $\text{poly}(C)$-close in parameter distance.*

We now show soundness of Algorithm 3, i.e., we show that no false candidate Gaussians will receive enough samples after the partial clustering procedure.

**Lemma 3.8.** *Partition $\mathcal{L}$ into groups $\mathcal{G}_1, \mathcal{G}_2, \ldots$, such that for all groups $\mathcal{G}_a$ and $\mathcal{G}_b$, there does not exist $(\mu_1, \Sigma_1) \in \mathcal{G}_a$ and $(\mu_2, \Sigma_2) \in \mathcal{G}_b$ with $a \neq b$ such that $d_{\mathsf{Param}}((\mu_1, \Sigma_1), (\mu_2, \Sigma_2)) \leq \Delta$. For each group $\mathcal{G}_i$ such that there does not exist $G_j = \mathcal{N}(\mu_j, \Sigma_j) \in \mathcal{D}$ and $(\mu, \Sigma) \in \mathcal{G}_i$ with $d_{\mathsf{Param}}(G_j, (\mu, \Sigma)) < Z_1\left(\frac{1}{\xi}\right)$, then with high probability, at most $\frac{\gamma}{200 \xi k}$ fraction of the points will be assigned to $\mathcal{G}_i$.*

We then show completeness of Algorithm 3, i.e.,we show that for each underlying Gaussian, there is a candidate Gaussian generated by the list-decoding procedure that is close in parameter distance and also receives a sufficient number of samples after the partial clustering procedure. Firstly, the existence of such a candidate Gaussian in the list output by the list-decoding procedure is given by Theorem 3.4. Thus it remains to show that it receives enough samples after the partial clustering procedure. We remark that this property only holds true because of the initial filtering of close pairs by Algorithm 1. If we do not perform such a filtering, then there could be many close candidate Gaussians that each cluster the samples generated from the true Gaussian, so that no individual cluster surpasses the threshold.

**Lemma 3.9.** *Partition $\mathcal{L}$ into groups $\mathcal{G}_1, \mathcal{G}_2, \ldots$, such that for all groups $\mathcal{G}_a$ and $\mathcal{G}_b$, there does not exist $(\mu_1, \Sigma_1) \in \mathcal{G}_a$ and $(\mu_2, \Sigma_2) \in \mathcal{G}_b$ with $a \neq b$ such that $d_{\mathsf{Param}}((\mu_1, \Sigma_1), (\mu_2, \Sigma_2)) \leq \Delta$. For each group $\mathcal{G}_i$ such that there exists $G_j = \mathcal{N}(\mu_j, \Sigma_j) \in \mathcal{D}$ and $(\mu, \Sigma) \in \mathcal{G}_i$ with $d_{\mathsf{Param}}(G_j, (\mu, \Sigma)) < \Delta$, then with high probability:*

*(1) At least $1 - \frac{\gamma}{400k}$ fraction of the points drawn from $G_j$ will be assigned to $\mathcal{G}_i$.*

*(2) At least $\frac{1}{2k}$ fraction of the points will be assigned to $\mathcal{G}_i$.*

Putting together the soundness and completeness claims in Lemma 3.8 and Lemma 3.9, respectively, we have the following statement:

**Lemma 3.10.** *Let $C_i$ be the set of points that are drawn from $G_i$ in $\mathcal{D}$. Let $\mathcal{G}_1, \mathcal{G}_2, \ldots$ be the groups of $\widetilde{\mathcal{P}}$. Then with probability $1 - \frac{1}{\mathrm{poly}(k)}$, there exists $\ell \in \mathbb{Z}$ such that for $Q_\ell = \mathcal{G}_\ell \cap C_i$, we have $|Q_\ell| > (1 - \gamma) \max(|\mathcal{G}_\ell|, |C_i|)$.*

As a result of soundness and completeness in Lemma 3.10, we have that with high probability, we are only left with $k$ candidate Gaussians that surpass the size threshold after the partial clustering procedure.

**Lemma 3.11.** *With probability $1 - \frac{1}{\mathrm{poly}(k)}$, Algorithm 3 outputs $k$ groups $\mathcal{G}_1, \ldots, \mathcal{G}_k$.*

Hence, we obtain our main result in Theorem 1.1.

Finally, we remark that our technique can similarly handle the case where a small but constant $\alpha$ fraction of the predicted lists of $\beta$ labels for each sample is incorrect, though we require the standard precision and recall assumption (Nguyen et al., 2023). Namely, we assume that the error rate is uniform across the clusters, i.e., there exists a sufficiently small error parameter $\alpha \in (0, 1)$ such that only an $\alpha$ fraction of all the samples generated from each Gaussian $i \in [k]$ are incorrect and furthermore the number of points that are given label $i$ in the predicted list is only a constant multiple of the true number of samples from $i$.

Observe that the only place the labels are being used are to apply the list-decoding algorithm of Ivkov & Kothari (2022) on the sets of points given each label $i \in [k]$. However, the statement of Theorem 3.4 permits a corruption rate $(1 - \alpha)$, which are currently fixing to handle additional points that are labeled $i$ from other clusters. In fact, Ivkov & Kothari (2022) study a more general corruption model, where an arbitrary $\alpha$ fraction of the input points may be altered adversarially. Thus, Theorem 3.4 can still handle the case where an $\alpha$ fraction of the samples generated by each Gaussian is deleted, which ultimately gives Theorem 1.2. Finally, we mention that there exists a more efficient algorithm to handle an $\alpha$ fraction of incorrect oracle responses when the list oracle returns a list of size $\beta = 1$. We defer discussion of this algorithm to Appendix B.

## 4 EMPIRICAL EVALUATIONS

To complement our theoretical guarantees, we implement a number of empirical evaluations on synthetic datasets to compare the performance of our learning-augmented algorithm with standard benchmarks.

**Experimental setup.** To generate our datasets, we define a mixture distribution $\mathcal{D}$ that consists of $k$ Gaussians, each with weight $\frac{1}{k}$. Although the exact value of the $k$ Gaussians differ across our experiments, it is important that these Gaussians are overlapping, since otherwise the resulting datasets are well-separated and can be easily solved by existing algorithms for learning mixtures of Gaussians. In fact, even existing algorithms for $k$-clustering problems can easily handle these cases. We then draw $n$ samples from the mixture distribution as the input to the Gaussian mixture model learning algorithms.

As a baseline, we run the popular Lloyd's algorithm for clustering (Lloyd, 1982). Lloyd's algorithm initializes $k$ centroids, which defines a partition on the input dataset, according to which of the $k$ centroids is closest to each point of the dataset. It then repeatedly computes the centroid of the points in each partition and then partitions the input dataset on these new centroids. finds the centroid of each set in the partition and then re-partitions the input. We first run Lloyd's algorithm for a small number of iterations to serve as our predictor. We then continue to run Lloyd's algorithm for a large number of iterations to server as the baseline.

For each point in the dataset, the list oracle outputs the two centroids of the predictor closest to the point. Due to the intricacies of the list-decoding algorithm of Ivkov & Kothari (2022), we run a scaled-down version of our algorithm. Furthermore, as the list decoding procedure identifies Frobenius-separated components, we focus on identifying mixtures of Gaussians that are either pairwise mean-separated or spectrally-separated. To that end, we use the list of centroids from the predictor to directly perform the partial clustering. That is, we first perform pruning to separate the clusters that are mean-separated. To handle the cluster that are spectrally-separated, we generate a random unit vector $u$ and an orthogonal unit vector $v$ and cluster the remaining points based on whether they are more aligned with $u$ or $v$. We then plot the resulting cluster induced by both the

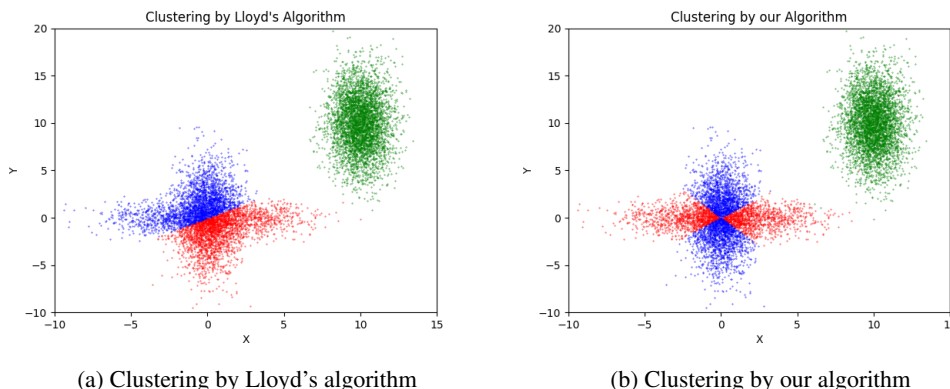

(a) Clustering by Lloyd's algorithm        (b) Clustering by our algorithm

Fig. 2: Clustering of $n = 10,000$ points generated from the Gaussian mixture model by Lloyd's algorithm (Figure 2a) and our learning-augmented algorithm (Figure 2b).

| Dataset size: | 5,000 | 10,000 | 15,000 | 20,000 | 25,000 |
|---|---|---|---|---|---|
| Lloyd's algorithm | 3606.51 | 7224.32 | 10832.35 | 14435.54 | 18032.83 |
| Learning-augmented algorithm | 4370.31 | 8740.69 | 13117.25 | 17485.38 | 21841.03 |
| Improvement (%) | 121.18 | 120.99 | 121.09 | 121.13 | 121.12 |

Table 1: Comparison of number of correctly classified points by Lloyd's algorithm and by our learning-augmented algorithm across various dataset sizes.

baseline algorithm and our learning-augmented algorithm. We also compare the number of points that are correctly labeled by both the baseline algorithm and our learning-augmented algorithm.

Our experiments were implemented using Python 3.12.6 on a Dell Precision 3620 workstation running Windows 10 on an Intel Core i7-6700 3.4GHz 4 Core Processor with 64GB DDR4 Memory and 512GB SSD.

**Results and discussion.** Our experimental results are aligned with our theoretical guarantees. Lloyd's algorithm repeatedly partitions the input space into a number of Voronoi cells, which is inherently well-suited for $k$-means clustering, but may not capture the more complex structural properties of mixtures of Gaussians, e.g., the individual Gaussians may be overlapping. Indeed, the classifications by Lloyd's algorithm results in linearly separated clusters as in Figure 2a. On the other hand, our learning-augmented algorithm produces clusters that capture the Gaussians in their key directions, c.f., Figure 2b. Specifically, our learning-augmented algorithm consistently correctly classifies a higher number of point than the learning-augmented algorithm, across the various parameters for the dataset size $n \in \{5000, 10000, 15000, 20000, 25000\}$. The classification rate of learning-augmented algorithm demonstrated an average improvement of roughly $120\%$ over Lloyd's algorithm across 100 iterations for each value of $n$. We summarize these results in Table 1.

## CONCLUSION

In this paper, we initiate the study of learning-augmented learning mixture models. Although well-known impossibility results state that any algorithm for learning a mixture of $k$ multivariate Gaussian distributions in $d$-dimensional space requires both runtime and sample complexity exponential in $k$, even if the Gaussians are somewhat well-separated, we show that with a reasonable (possibly erroneous) list oracle, we can estimate the mixture up to total variation distance $\tilde{\mathcal{O}}(\varepsilon)$ using $k \cdot \text{poly}\left(d, \log k, \frac{1}{\varepsilon}\right)$ samples from the GMM, i.e., only a polynomial number of samples, thereby breaking the computational limitations of traditional algorithms that are unable to incorporate machine learning advice. It is our hope that our work is an important step towards a greater understanding of learning theory that incorporates possibly erroneous input into better provable guarantees.

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

# A  PRELIMINARIES

We write $[n] := \{1, 2, \ldots, n\}$ for a positive integer $n \geq 1$. We use $\mathrm{poly}(n)$ to denote a fixed polynomial in $n$ that can be determined by adjusting constants appropriately. We say that an event occurs with high probability if it occurs with probability $1 - \frac{1}{\mathrm{poly}(n)}$ for the relevant variable $n$, e.g., if a union bound over $k$ clusters is necessary, then we say an event occurs with high probability if it occurs with probability $1 - \frac{1}{\mathrm{poly}(k)}$.

**Definition A.1** (Total variation distance). *For probability density functions $p(x)$ and $q(x)$ over a space $\Omega$, we write the total variation distance between $p$ and $q$ as $d_{\mathsf{TV}}(p, q) := \frac{1}{2} \int_{s \in \Omega} |p(x) - q(x)| \, dx$.*

For the purposes of concentration inequalities, we recall the following standard multiplicative form of Chernoff bounds.

**Theorem A.2** (Chernoff bounds). *Let $X_1, \ldots, X_n$ be independent binary random variables, i.e., $X_i \in \{0, 1\}$ for all $i \in [n]$. Let $X = X_1 + \ldots + X_n$ and $\mu = \mathbb{E}[X]$. Then for any $\delta \in (0, 1)$,*

$$\mathbf{Pr}\left[|X - \mu| \geq \delta \mu\right] \leq 2 \exp\left(-\delta^2 \mu / 3\right).$$

# B  LABEL ORACLES

In this section, we consider learning a mixture $\mathcal{D}$ of $k$ Gaussians given a *label oracle*, which provides a label for each sample, i.e., the index of the corresponding Gaussian, up to some error $\alpha$. Specifically, if $x \sim \mathcal{D}$ is drawn from Gaussian $G_i = (\mu_i, \Sigma_i)$, then informally, the oracle will provide the label $i$ on query $x$ with probability at least $(1 - \alpha)$. More formally, the error of the oracle needs to respect precision and recall of each cluster as follows:

**Definition B.1** (Label oracle). *Given a set $X$ of $n$ samples $x_1, \ldots, x_n \subset \mathbb{R}^d$ from a mixture $\mathcal{D}$ of $k$ Gaussians $G_1, \ldots, G_k$, let $C_1, \ldots, C_k$ be the partition of $X$ such that $x_i \in C_j$ if $x_i$ is drawn from $G_j$ and let $m_j = |G_j|$. Then a label oracle with error rate $\alpha$ partitions the points into clusters $P_1, \ldots, P_k$ such that for $Q_j = P_j \cap C_j$, we have $|Q_j| > (1 - \alpha) \max(|P_j|, |C_j|)$ for all $j \in [k]$, where $\alpha < \frac{1}{2}$ is an error rate for the predictor.*

We remark that a label oracle is a simple heuristic to generate. Consider some clustering of a previous set of samples from the dataset. The clustering can be used to train a model that will provide a label corresponding to one of the clusters, for each query point. In the case the data is well-separated, such a clustering will generally provide good accuracy over samples from the mixture and thus serve as a good label oracle.

We also emphasize that only a certain fraction of the labels are required to be correct, and the remaining labels can be arbitrarily incorrect. In particular, the incorrect labels can be adversarially incorrect. We show that learning a mixture of $k$ Gaussians up to arbitrary total variation distance $\tilde{\mathcal{O}}(\varepsilon)$ can be achieved in polynomial runtime and sample complexity, with the aid of a label oracle. As a warm-up, we first show in Appendix B.1 that a label oracle with error rate $\alpha \leq \varepsilon$, in addition to $k \cdot \mathrm{poly}\left(d, \log k, \frac{1}{\varepsilon}\right)$ samples from the mixture $\mathcal{D}$ of $k$ Gaussians, can be used to output a mixture $\mathcal{D}'$ of $k$ Gaussians such that $d_{\mathsf{TV}}(\mathcal{D}, \mathcal{D}') \leq \tilde{\mathcal{O}}(\varepsilon)$.

**Theorem B.2.** *Given a label oracle with error rate $\alpha \leq \varepsilon$, there exists an algorithm that takes $n = k \cdot \mathrm{poly}\left(d, \log k, \frac{1}{\varepsilon}\right)$ samples from any $d$-dimensional mixture of Gaussians $\sum_{i \in [k]} w_i G_i$ for all $i \in [k]$, runs in time $k \cdot \mathrm{poly}(n)$, and returns $k$ hypothesis Gaussians $H_1, \ldots, H_k$ and weights $u_1, \ldots, u_k$ such that with high probability,*

$$d_{\mathsf{TV}}\left(\sum_{i \in [k]} w_i G_i, \sum_{i \in [k]} u_i H_i\right) \leq \tilde{\mathcal{O}}(\varepsilon).$$

We remark that the naïve algorithm of simply outputting the empirical mean and covariances of the clusters induced by the label oracle cannot give the guarantees of Theorem B.2, even as the error rate becomes arbitrarily small.

Nevertheless, we can achieve Theorem B.2 through the following simple observation. Because the fraction of corrupted labels is rather small, they can only substantially affect the empirical mean and covariances of the set of points with each label if they are quite "far" from the true distribution. For example, the points that are falsely labeled $i$ must be quite far in the likelihood function from the "normal" distribution of the Gaussian $G_i = (\mu_i, \Sigma_i)$ for the empirical mean and covariance of the points labeled $i$ to have large total variation distance from $G_i$. In this case, we can observe that these samples are far from the other samples labeled $i$ and prune them away before computing the empirical mean and covariance of the remaining points. It is also possible for the corrupted labels to be close to $G_i$ in the likelihood function, but then the empirical mean and covariance are only perturbed by a small amount.

Note that the above argument crucially relies on the label oracle having error rate $\alpha \leq \varepsilon$, where $\varepsilon$ is the target total variation distance between the output mixture distribution and the true mixture distribution. Surprisingly, we then show in Section B.2 that the target total variation distance can be obtained even when the error rate of the oracle is a small constant error rate $\alpha = \Omega(1)$, independent of the target total variation distance accuracy parameter $\varepsilon$.

**Theorem B.3.** *Given an $\alpha$-error oracle for a sufficiently small constant $\alpha = \mathcal{O}(1)$, there exists an algorithm that takes $n = \text{poly}(dk/\varepsilon)$ samples from any $d$-dimensional mixture of Gaussians $\sum_{i \in [k]} w_i G_i$ for all $i \in [k]$, runs in time $k \cdot \text{poly}(n)$, and returns $k$ hypothesis Gaussians $H_1, \ldots, H_k$ and weights $u_1, \ldots, u_k$ such that with high probability,*

$$d_{\text{TV}}\left(\sum_{i \in [k]} w_i G_i, \sum_{i \in [k]} u_i H_i\right) \leq \tilde{\mathcal{O}}(\varepsilon).$$

Theorem B.3 uses an observation by Diakonikolas et al. (2020) that it suffices to recover a large constant fraction of the samples for each cluster. Then using an additional set of $\text{poly}(dk/\varepsilon)$ samples, we can obtain a mixture distribution $\mathcal{D}'$ close to true mixture distribution $\mathcal{D}$.

### B.1 WARM-UP: SMALL ERROR LABEL ORACLES

We first describe a simple algorithm that uses a label oracle with error rate $\alpha \leq \varepsilon$ and polynomial runtime and number of samples to learning a mixture of $k$ Gaussians up to arbitrary total variation distance $\tilde{\mathcal{O}}(\varepsilon)$. The main component of the algorithm is the following subroutine by Diakonikolas et al. (2020) that takes a samples from a mixture of Gaussians and corrupts an $\varepsilon$ fraction of the samples to any adversarial set of points.

**Theorem B.4** (Theorem 1.2 in Diakonikolas et al. (2020)). *For every $w_{\min}$, there are functions $F(w_{\min})$, $f(w_{\min})$ such that there is an algorithm PRUNEERR that takes $n = d^{F(w_{\min})}/\text{poly}(\varepsilon)$ $\varepsilon$-corrupted samples from any $d$-dimensional mixture of Gaussians $\sum_{i \in [k]} w_i G_i$ with $w_i \geq w_{\min}$ for all $i \in [k]$, runs in time $n^{F(w_{\min})}$, and returns $k$ hypothesis Gaussians $H_1, \ldots, H_k$ and weights $u_1, \ldots, u_k$ such that with high probability, there exists a permutation $\pi : [k] \rightarrow [k]$ for which $\max_i d_{\text{TV}}(G_i, H_{\pi(i)}) \leq \tilde{\mathcal{O}}(\varepsilon)$ and $\sum_{i \in [k]} |w_i - u_{\pi(i)}| \leq \mathcal{O}(\varepsilon)$, provided $\varepsilon \leq f(w_{\min})$ and $\min_{i \neq j} d_{\text{TV}}(G_i, G_j) \geq 1 - f(w_{\min})$.*

Here, $\varepsilon$-corrupted simply means that $\varepsilon$ fraction of the samples may be adversarially generated.

The challenge is that Theorem B.4 uses runtime and sample complexity exponential in the inverse weight of the Gaussians. Thus even in a uniform mixture, the runtime and sample complexity is exponential in $k$.

Nevertheless, our algorithm for learning the mixture of Gaussians is simple. We sample a polynomial number of points and partition the points by the labels given from the oracle. For each $i \in [k]$, we then run the robust Gaussian recovery algorithm each set of points labeled $i$, thus bypassing the hard case of Theorem B.4 since there is only one true Gaussian $G_i$ that generates the majority of the points. We give the full algorithm in Algorithm 4.

We justify the full guarantees of Algorithm 4.

**Theorem B.2.** *Given a label oracle with error rate $\alpha \leq \varepsilon$, there exists an algorithm that takes $n = k \cdot \text{poly}\left(d, \log k, \frac{1}{\varepsilon}\right)$ samples from any $d$-dimensional mixture of Gaussians $\sum_{i \in [k]} w_i G_i$ for*

---

**Algorithm 4** Learning-augmented algorithm for learning mixture of $k$-Gaussians, small corruption rate

---

**Input:** Corruption parameter $\alpha > 0$, set $X$ of $k \log k \cdot \text{poly}(d/\varepsilon)$ samples $s_1, \ldots, s_n \subset \mathbb{R}^d$ from a mixture $\mathcal{D}$ of $k$ Gaussians $G_1, \ldots, G_k$

**Output:** $\widetilde{\mathcal{D}}$ with $d_{\mathsf{TV}}(\mathcal{D}, \widetilde{\mathcal{D}}) \leq \tilde{\mathcal{O}}(\varepsilon)$

1: **for** $i = 1$ to $i = k$ **do**
2:     Let $P_i$ be the set of points labeled $i$
3:     $u_i \leftarrow \frac{|P_i|}{n}$
4:     Let $H_i$ be the output of PRUNEERR on $P_i$ with corruption $\alpha$ and $w_{\min} = 1$     ▷Theorem B.4
5: **end for**
6: Output $\widetilde{\mathcal{D}} = \{(u_i, H_i)\}_{i \in [k]}$

---

*all $i \in [k]$, runs in time $k \cdot \text{poly}(n)$, and returns $k$ hypothesis Gaussians $H_1, \ldots, H_k$ and weights $u_1, \ldots, u_k$ such that with high probability,*

$$d_{\mathsf{TV}}\left(\sum_{i \in [k]} w_i G_i, \sum_{i \in [k]} u_i H_i\right) \leq \tilde{\mathcal{O}}(\varepsilon).$$

*Proof.* Without loss of generality, suppose $G_i$ corresponds to $P_i$ and hence $H_i$ for each $i \in [k]$. We first claim that for any $i \in [k]$, we have $|u_i - w_i| \leq \mathcal{O}(\varepsilon)$. Note that for each sample, the probability that is is drawn from $G_i$ is $w_i$. Thus by linearity of expectation, the expected number of samples that are drawn from $G_i$ across $n$ samples is $w_i n$. Then for $n = k \log k \cdot \text{poly}(d/\varepsilon)$, we have that the total number of samples drawn from $G_i$ is at most $\mathcal{O}(\varepsilon) \cdot w_i n$ away from its expectation with high probability. Since an additional $\alpha \leq \varepsilon$ fraction of these samples are corrupted, then it follows that $|u_i - w_i| \leq \mathcal{O}(\varepsilon)$.

Moreover, each set $P_i$ is a set of $\varepsilon$-corrupted samples from a $d$-dimensional Gaussian. Thus by Theorem B.4, for the output $H_i$ of PRUNEERR on $P_i$ with corruption $\alpha$ and $w_{\min} = 1$, we have that $d_{\mathsf{TV}}(G_i, H_i) \leq \tilde{\mathcal{O}}(\varepsilon)$ with high probability. By a union bound, we have $|u_i - w_i| \leq \mathcal{O}(\varepsilon)$ and $d_{\mathsf{TV}}(G_i, H_i) \leq \tilde{\mathcal{O}}(\varepsilon)$ for all $i \in [k]$. Therefore, we have that

$$d_{\mathsf{TV}}\left(\sum_{i \in [k]} w_i G_i, \sum_{i \in [k]} u_i H_i\right) \leq \tilde{\mathcal{O}}(\varepsilon).$$

$\square$

### B.2 CONSTANT ERROR LABEL ORACLES

We describe how to learn a GMM to a target total variation distance parameter $\varepsilon$ if the label oracle has error rate independent of $\varepsilon$. Note that the framework in Appendix B.1 cannot be applied because even restricted to only the samples labeled some fixed $i \in [k]$, at least a constant fraction of the samples may be corrupted. Fortunately, there exists an algorithm that can recover the true Gaussians, given a constant fraction of corruptions.

**Theorem B.5** (Proposition 8.3 and Proposition 8.4 in (Diakonikolas et al., 2020)). *Let $X = \sum_{i \in [k]} w_i G_i$ be a mixture of $k$ Gaussians with $-\log(1 - d_{\mathsf{TV}}(G_i, G_j) = \Omega(\log(k/\varepsilon))$ for all $i \neq j$. Let $X'$ be an $\varepsilon$-corrupted version of $X$ and $n = \text{poly}(dk/\varepsilon)$. Let $S$ be a set of $n$ random samples from $X'$ and let $T_1, \ldots, T_k \subset S$ be sets of samples so that for some sufficiently small constant $\gamma > 0$, if $S_i$ is the set of samples in $S$ that were drawn from $G_i$, then $|T_i \cap S_i| \geq (1 - \gamma) \min(|T_i|, |S_i|)$ for all $i \in [k]$. There exists an algorithm ROBUSTGAUSSIANS that takes input $S, T_1, \ldots, T_k$, and an additional set of $n$ independent samples from $X'$ and outputs a set of weights $u_1, \ldots, u_k$ such that $\sum_{i=1}^{k} |u_i - w_i| = \mathcal{O}(\varepsilon)$ and a list of Gaussians $H_1, \ldots, H_k$ so that $\sum_{i \in [k]} w_i d_{\mathsf{TV}}(H_i, G_i) \leq \mathcal{O}(\varepsilon \log \frac{1}{\varepsilon})$. The algorithm uses $\text{poly}(n)$ runtime.*

We again note that running Theorem B.5 alone on the input dataset does not suffice, as it would result in runtime and sample complexity that is exponential in $k$. The label oracle is implicitly performing

heavy work in reducing the input from a mixture of $k$ Gaussians to $k$ instances of a mixture of a single Gaussian and a number of corrupted samples. We give the full algorithm in Algorithm 5.

---

**Algorithm 5** Learning-augmented algorithm for learning mixture of $k$-Gaussians, constant corruption rate

---

**Input:** Corruption parameter $\alpha > 0$, set $X$ of $k \cdot \mathrm{poly}(d/\varepsilon)$ samples $s_1, \ldots, s_n \subset \mathbb{R}^d$ from a mixture $\mathcal{D}$ of $k$ Gaussians $G_1, \ldots, G_k$
**Output:** $\widetilde{\mathcal{D}}$ with $d_{\mathsf{TV}}(\mathcal{D}, \widetilde{\mathcal{D}}) \leq \tilde{\mathcal{O}}(\varepsilon)$
  1: $\mathcal{P} \leftarrow \emptyset$
  2: **for** $i = 1$ to $i = k$ **do**
  3:   Let $P_i$ be the set of points labeled $i$
  4:   $\mathcal{P} \leftarrow \mathcal{P} \cup P_i$
  5: **end for**
  6: Let $\widetilde{\mathcal{D}} = \{(u_i, H_i)\}_{i \in [k]}$ be the output of ROBUSTGAUSSIANS on $\mathcal{P}$ with accuracy $\varepsilon$
      ▷Theorem B.5
  7: Output $\widetilde{\mathcal{D}}$

---

We now give the full guarantees of Algorithm 5.

**Proof of Theorem B.3:** Consider Algorithm 5 and for each $i \in [k]$, let $P_i$ be the set of points labeled $i$. Moreover, for each $i \in [k]$, let $X_i$ be the set of points in the sample set $X$ that are drawn from $G_i$. Note that for a sufficiently small constant $\alpha < \gamma$, we have

$$|P_i \cap X_i| > (1 - \gamma) \max(|P_i|, |X_i|),$$

thus satisfying the conditions of Theorem B.5. Hence by Theorem B.5, the output $\widetilde{\mathcal{D}} = \{(u_i, H_i)\}_{i \in [k]}$ satisfies

$$d_{\mathsf{TV}}\left(\sum_{i \in [k]} w_i G_i, \sum_{i \in [k]} u_i H_i\right) \leq \tilde{\mathcal{O}}(\varepsilon).$$

$\square$

## C MISSING PROOFS FROM SECTION 3

In this section, we supply the proofs missing from Section 3.

### C.1 PARAMETER DISTANCE SEPARATION TO TVD SEPARATION

In this section, we show Lemma 3.3, which states that two Gaussian distributions with parameter distance lower bounded by $\Delta$ must also have their total variation distance lower bounded by a fixed function of $\Delta$. We prove Lemma 3.3 by performing casework, separately considering Gaussians pairs $(\mu_1, \Sigma_1)$ and $(\mu_2, \Sigma_2)$ that are mean-separated, Frobenius-separated, and spectrally-separated.

#### C.1.1 MEAN-SEPARATION

The proof lower bounding the total variation distance of two Gaussians that are mean-separated is simple. It follows quite naturally from the definition of mean-separation as follows.

**Lemma C.1.** *Suppose there exists $v \in \mathbb{R}^d$, such that $\langle \mu_1 - \mu_2, v \rangle^2 > \Delta v^\top (\Sigma_1 + \Sigma_2) v$. Then $d_{\mathsf{TV}}(\mathcal{N}(\mu_1, \Sigma_1), \mathcal{N}(\mu_2, \Sigma_2)) \geq 1 - \exp(-\mathcal{O}(\Delta/\log \Delta))$.*

*Proof.* Let $x \sim \mathcal{N}(\mu_1, \Sigma_1)$, so that $\langle x, v \rangle \sim \mathcal{N}(\langle \mu_1, v \rangle, v^\top \Sigma_1 v)$. Note that

$$\left\langle \frac{\mu_1 - \mu_2}{2}, v \right\rangle > \frac{\Delta}{2} \cdot v^\top (\Sigma_1 + \Sigma_2) v.$$

Thus,

$$\mathbf{Pr}\left[\langle x - \mu_1, v \rangle \geq \left\langle \frac{\mu_1 - \mu_2}{2}, v \right\rangle\right] \leq \exp(-\mathcal{O}(\Delta^2)).$$

By similar reasoning, we have that for $y \sim \mathcal{N}(\mu_2, \Sigma_2)$,

$$\mathbf{Pr}\left[\langle y - \mu_2, v \rangle \geq \left\langle \frac{\mu_1 - \mu_2}{2}, v \right\rangle\right] \leq \exp(-\mathcal{O}(\Delta^2)).$$

Therefore, we certainly have $d_{\mathsf{TV}}(\mathcal{N}(\mu_1, \Sigma_1), \mathcal{N}(\mu_2, \Sigma_2)) \geq 1 - \exp(-\mathcal{O}(\Delta/\log\Delta))$. ☐

### C.1.2 FROBENIUS-SEPARATION

The analysis for the total variation distance of two Gaussians that are Frobenius-separated is more challenging. For two probability distributions $p$ and $q$, we use $V(p, q) = \int \min(dp, dq)$ to denote the overlap of the distributions, so that $d_{\mathsf{TV}}(p, q) = 1 - V(p, q)$. We define $h(p, q) = -\log\left(\int \sqrt{dpdq}\right)$. Then we have the following statement relating the overlap of the distributions with $h(p, q)$.

**Lemma C.2.** $V(p, q) \leq \exp(-h(p, q))$

*Proof.* By definition of $V(p, q)$ and $h(p, q)$, we have

$$V(p, q) = \int \min(dp, dq) \leq \int \sqrt{dpdq} = \exp(-h(p, q)).$$

☐

For weighted Gaussians $G_1 = w_1 \cdot \mathcal{N}(\mu_1, \Sigma_1)$ and $G_2 = w_2 \cdot \mathcal{N}(\mu_2, \Sigma_2)$, we define

$$h_\Sigma(G_1, G_2) := h(\mathcal{N}(0, \Sigma_1), \mathcal{N}(0, \Sigma_2)).$$

Then we have:

**Lemma C.3.** *For weighted Gaussians $G_1 = w_1 \cdot \mathcal{N}(\mu_1, \Sigma_1)$ and $G_2 = w_2 \cdot \mathcal{N}(\mu_2, \Sigma_2)$, we have*

$$h(G_1, G_2) \geq h_\Sigma(G_1, G_2).$$

Next, we recall the following statement from Diakonikolas et al. (2017) that relates $h_\Sigma(G_1, G_2)$ to the eigenvalues of $\Sigma_2^{-1/2}\Sigma_1\Sigma_2^{-1/2}$.

**Lemma C.4** (Lemma B.4 in (Diakonikolas et al., 2017))**.** *For weighted Gaussians $G_1 = w_1 \cdot \mathcal{N}(\mu_1, \Sigma_1)$ and $G_2 = w_2 \cdot \mathcal{N}(\mu_2, \Sigma_2)$, let $\lambda_1, \ldots, \lambda_d$ be the eigenvalues of $\Sigma_2^{-1/2}\Sigma_1\Sigma_2^{-1/2}$. Then*

$$h_\Sigma(G_1, G_2) = \Theta\left(\sum_\lambda \min(|\log(\lambda)|, |\log(\lambda)|^2)\right).$$

The following statement is shown in both the proof of Lemma B.4 in (Diakonikolas et al., 2017) and the proof of Lemma 3.6 in (Diakonikolas et al., 2020).

**Lemma C.5.** *(Diakonikolas et al., 2017; 2020) Let $G_1 = w_1 \cdot \mathcal{N}(\mu_1, \Sigma_1)$ and $G_2 = w_2 \cdot \mathcal{N}(\mu_2, \Sigma_2)$ be weighted Gaussians that are Frobenius-separated, i.e., $\|I_d - \Sigma_2^{-1/2}\Sigma_1\Sigma_2^{-1/2}\|_F^2 \geq \Omega\left(\frac{\log 1/\varepsilon}{\log\log 1/\varepsilon}\right)$ but are not mean-separated or spectral separated. Let $\lambda_1, \ldots, \lambda_d$ be the eigenvalues of $\Sigma_2^{-1/2}\Sigma_1\Sigma_2^{-1/2}$. Then*

$$\sum_\lambda \min(|\log(\lambda)|, |\log(\lambda)|^2) \geq \Omega\left(\log\frac{1}{\varepsilon}\right).$$

Putting these statements together, we have:

**Corollary C.6.** *Let $G_1 = w_1 \cdot \mathcal{N}(\mu_1, \Sigma_1)$ and $G_2 = w_2 \cdot \mathcal{N}(\mu_2, \Sigma_2)$ be weighted Gaussians that are $C$-Frobenius-separated. Then $d_{\mathsf{TV}}(G_1, G_2) \geq 1 - 2^{-\mathcal{O}(C/\log C)}$.*

### C.1.3 SPECTRAL-SEPARATION

Finally, we analyze the total variation distance of two Gaussians that are spectrally-separated.

**Lemma C.7.** *Suppose the mean-covariance pair $(\mu_1, \Sigma_1)$ and $(\mu_2, \Sigma_2)$ satisfies Mahalanobis mean closeness, i.e., for all $v \in \mathbb{R}^d$, we have $\langle \mu_1 - \mu_2, v \rangle^2 \leq \Delta v^\top (\Sigma_1 + \Sigma_2) v$. Moreover suppose are spectrally-separated, i.e., there exists $v \in \mathbb{R}^d$, such that $\frac{1}{\Delta} v^\top \Sigma_2 v \leq v^\top \Sigma_1 v > \Delta v^\top \Sigma_2 v$. Then $d_{\mathsf{TV}}(\mathcal{N}(\mu_1, \Sigma_1), \mathcal{N}(\mu_2, \Sigma_2)) \geq 1 - \Delta^{-\mathcal{O}(1)}$.*

*Proof.* Let $V_i = v^\top \widehat{\Sigma_i} v$ and $V_j = v^\top \Sigma_j v$ and suppose without loss of generality that $V_i < V_j$. Consider $x \sim \mathcal{N}(\mu_i, \Sigma_i)$. By Fact 2.2, we have that the distribution of $\mathrm{Proj}(x - \mu_i, v)$ is the distribution $\mathcal{N}(0, v^\top \Sigma_i v)$. Then

$$\mathbf{Pr}\left[ \| \mathrm{Proj}(x - \mu_i, v) \|^2 \geq \sqrt{\Delta} \cdot v^\top \widehat{\Sigma_i} v \right] < \exp(-\mathcal{O}(\Delta)).$$

On the other hand, for $y \sim \mathcal{N}(\mu_j, \Sigma_j)$, we have that the distribution of $\mathrm{Proj}(y - \mu_j, v)$ is the distribution $\mathcal{N}(0, v^\top \Sigma_j v)$. Thus,

$$\mathbf{Pr}\left[ \| \mathrm{Proj}(y - \mu_j, v) \|^2 \leq \frac{1}{\sqrt{\Delta}} \cdot v^\top \widehat{\Sigma_j} v \right] < \Delta^{-\mathcal{O}(1)}.$$

Thus we have $d_{\mathsf{TV}}(\mathcal{N}(\mu_1, \Sigma_1), \mathcal{N}(\mu_2, \Sigma_2)) \geq 1 - \Delta^{-\mathcal{O}(1)}$. $\qquad\square$

### C.1.4 PUTTING THINGS TOGETHER

By the definition of parameter distance, a large parameter distance between a pair of Gaussians implies that they are mean-separated, Frobenius-separated, or spectrally-separated. Thus, putting together the separate results from Lemma C.1, Corollary C.6, Lemma C.7, we have:

**Lemma 3.3.** *For $\Delta \geq 1$, suppose $\mu_1, \mu_2$ and $\Sigma_1, \Sigma_2$ satisfy $d_{\mathsf{Param}}((\mu_1, \Sigma_1), (\mu_2, \Sigma_2)) > \Delta$. Then $d_{\mathsf{TV}}(\mathcal{N}(\mu_1, \Sigma_1), \mathcal{N}(\mu_2, \Sigma_2)) \geq 1 - \frac{1}{\mathrm{poly}(\Delta)}$.*

*Proof.* Recall that the definition of parameter distance separation implying that the mean-covariance pairs are either mean-separated, Frobenius-separated, or spectrally-separated. If the mean-covariance pairs are mean-separated, then Lemma C.1 shows they are also TVD-separated. If the mean-covariance pairs are spectrally-separated but not mean-separated, then Lemma C.7 shows they are also TVD-separated. Finally, if the mean-covariance pairs are Frobenius-separated, then Corollary C.6 shows they are also TVD-separated. $\qquad\square$

### C.2 REMAINING MISSING PROOFS FROM SECTION 3

**Observation 3.5.** *[Well-separatedness of $\mathcal{L}$] Let $\mathcal{L} = \{(\mu_i, \Sigma_i)\}$ be the output of MERGECLOSEPAIRS. Then for each $i \neq j$, we have $d_{\mathsf{Param}}((\mu_i, \Sigma_i), (\mu_j, \Sigma_j)) > \Delta$.*

*Proof.* Note that if $d_{\mathsf{Param}}((\mu_i, \Sigma_i), (\mu_j, \Sigma_j)) \leq \Delta$, then either $(\mu_i, \Sigma_i)$ or $(\mu_j, \Sigma_j)$ would have been deleted from $\mathcal{L}$ by algorithm MERGECLOSEPAIRS. $\qquad\square$

**Observation 3.6.** *[Partial clustering of $X$] Let $x \in X$ and let $\mathcal{P} = \{P_1, \ldots, P_{|\mathcal{P}|}\}$ be the output of PARTIALCLUSTER$(X, \mathcal{L})$ for any $\mathcal{L}$. Then there exists exactly one index $i \in [|\mathcal{P}|]$ such that $x \in P_i$.*

*Proof.* Observe that for each $x \in X$, PARTIALCLUSTER computes $j = \mathrm{argmax}_{i \in [\mathcal{K}]} p_i(x)$, breaking possible ties arbitrarily and adds $x$ only to $\mathcal{P}_j$. $\qquad\square$

To show that high variation distance implies the success of the maximum likelihood estimator, we use the following observation from Diakonikolas et al. (2020), e.g., inherently in the proof of Proposition 8.3.

**Lemma C.8.** *Diakonikolas et al. (2020) Suppose $G_1$ and $G_2$ be two Gaussian distributions such that $d_{\mathsf{TV}}(G_1, G_2) \geq 1 - \varepsilon$. Then the maximum likelihood estimator for determining will fail to correctly classify a sample $x \sim \mathcal{G}_i$ for $i \in \{1, 2\}$ with probability $\mathcal{O}(\varepsilon)$.*

**Lemma 3.8.** *Partition $\mathcal{L}$ into groups $\mathcal{G}_1, \mathcal{G}_2, \ldots$, such that for all groups $\mathcal{G}_a$ and $\mathcal{G}_b$, there does not exist $(\mu_1, \Sigma_1) \in \mathcal{G}_a$ and $(\mu_2, \Sigma_2) \in \mathcal{G}_b$ with $a \neq b$ such that $d_{\mathsf{Param}}((\mu_1, \Sigma_1), (\mu_2, \Sigma_2)) \leq \Delta$. For each group $\mathcal{G}_i$ such that there does not exist $G_j = \mathcal{N}(\mu_j, \Sigma_j) \in \mathcal{D}$ and $(\mu, \Sigma) \in \mathcal{G}_i$ with $d_{\mathsf{Param}}(G_j, (\mu, \Sigma)) < Z_1\left(\frac{1}{\xi}\right)$, then with high probability, at most $\frac{\gamma}{200\xi k}$ fraction of the points will be assigned to $\mathcal{G}_i$.*

*Proof.* We have two cases. Either $\mathcal{G}_i$ has parameter distance at least $\Omega\left(Z_1\left(\frac{1}{\xi}\right)\left(\frac{\xi k}{\gamma}\right)^{\zeta+1}\right)$ from $G_c$ for all $c \in [k]$ or there exists $c \in [k]$ and $(\mu, \Sigma) \in \mathcal{G}_i$ such that $d_{\mathsf{Param}}(G_c, (\mu, \Sigma)) = \mathcal{O}\left(Z_1\left(\frac{1}{\xi}\right)\left(\frac{\xi k}{\gamma}\right)^{\zeta+1}\right)$. Consider the first case, so that $\mathcal{G}_i$ has parameter distance at least $\Omega\left(Z_1\left(\frac{1}{\xi}\right)\left(\frac{\xi k}{\gamma}\right)^{\zeta+1}\right)$ from $G_c$ for all $c \in [k]$. On the other hand, each point is drawn from some distribution that has parameter distance at most $\mathsf{poly}\left(Z_1\left(\frac{1}{\xi}\right)\right)$ from some $(\mu, \Sigma) \in \mathcal{G}_i$. By Lemma 3.2, Lemma 3.3, and Lemma C.8, each point drawn from the mixture will be assigned to $\mathcal{G}_i$ with probability at most $\mathsf{poly}\left(-Z_1\left(\frac{1}{\xi}\right)\left(\frac{\xi k}{\gamma}\right)^{\zeta+1}\right)$. Thus, the total fraction of points assigned to $\mathcal{G}_i$ will be at most $\frac{\gamma}{200\xi k}$ with high probability.

In the second case, let $c \in [k]$ be fixed so that there exists $(\mu, \Sigma) \in \mathcal{G}_i$ such that $d_{\mathsf{Param}}(G_c, (\mu, \Sigma)) = \mathcal{O}\left(Z_1\left(\frac{1}{\xi}\right)\left(\frac{\xi k}{\gamma}\right)^{\zeta+1}\right)$. By assumption, that there does not exist $(\mu', \Sigma') \in \mathcal{G}_i$ with $d_{\mathsf{Param}}(G_c, (\mu', \Sigma')) < Z_1\left(\frac{1}{\xi}\right)$. Then by the contrapositive of Lemma 3.7 and the setting of $\Delta = Z_1\left(\frac{1}{\xi}\right)\left(\frac{\xi k}{\gamma}\right)^{\zeta}$ for a sufficiently high constant $\zeta$, we have that $G_c$ and $\mathcal{G}_i$ must be $D$-separated, for $D = Z_1\left(\frac{1}{\xi}\right)\left(\frac{\xi k}{\gamma}\right)^3$. Moreover, there exists $\mathcal{G}_j$ such that there exists $(\widetilde{\mu}, \widetilde{\Sigma}) \in \mathcal{G}_j$ with $d_{\mathsf{Param}}(G_c, (\widetilde{\mu}, \widetilde{\Sigma})) < Z_1\left(\frac{1}{\xi}\right)$. Thus by Lemma 3.2 and Lemma 3.3, each point drawn $x \sim G_c$, we have that the maximum likelihood function assigns $x$ to $\mathcal{G}_i$ with probability at most $\frac{\gamma}{400\xi}$. We have that each point is drawn from $G_c$ with probability $\frac{1}{k}$ and so with high probability, at most $\frac{\gamma}{300\xi k}$ fraction of the points will be drawn from $G_c$ and assigned to $\mathcal{G}_i$.

It thus remains to consider the points drawn from other clusters that are assigned to $\mathcal{G}_i$. For each point $x \sim G_{c'}$ with $c \neq c'$, we have that $x$ is assigned to $\mathcal{G}_i$ with probability at most $\mathsf{poly}\left(-Z_1\left(\frac{1}{\xi}\right)\left(\frac{\xi k}{\gamma}\right)^{\zeta+1}\right)$, by again applying Lemma 3.2 and Lemma 3.3. Therefore, the total fraction of points assigned to $\mathcal{G}_i$ will be at most $\frac{\gamma}{200\xi k}$ with high probability. $\qquad\square$

**Lemma 3.9.** *Partition $\mathcal{L}$ into groups $\mathcal{G}_1, \mathcal{G}_2, \ldots$, such that for all groups $\mathcal{G}_a$ and $\mathcal{G}_b$, there does not exist $(\mu_1, \Sigma_1) \in \mathcal{G}_a$ and $(\mu_2, \Sigma_2) \in \mathcal{G}_b$ with $a \neq b$ such that $d_{\mathsf{Param}}((\mu_1, \Sigma_1), (\mu_2, \Sigma_2)) \leq \Delta$. For each group $\mathcal{G}_i$ such that there exists $G_j = \mathcal{N}(\mu_j, \Sigma_j) \in \mathcal{D}$ and $(\mu, \Sigma) \in \mathcal{G}_i$ with $d_{\mathsf{Param}}(G_j, (\mu, \Sigma)) < \Delta$, then with high probability:*

*(1) At least $1 - \frac{\gamma}{400k}$ fraction of the points drawn from $G_j$ will be assigned to $\mathcal{G}_i$.*

*(2) At least $\frac{1}{2k}$ fraction of the points will be assigned to $\mathcal{G}_i$.*

*Proof.* Observe that there exists $(\mu, \Sigma) \in \mathcal{G}_i$ with $d_{\mathsf{Param}}(G_j, (\mu, \Sigma)) < Z_1\left(\frac{1}{\xi}\right)$. On the other hand, $d_{\mathsf{Param}}((\mu, \Sigma), (\mu', \Sigma')) > \Delta$ for any $(\mu', \Sigma_1) \in \mathcal{G}_a$ with $a \neq i$, where $\Delta = Z_1\left(\frac{1}{\xi}\right)\left(\frac{\xi k}{\gamma}\right)^{\zeta}$. Then for each point drawn $x \sim G_j$, we have that the maximum likelihood function assigns $x$ to $\mathcal{G}_a$ with probability at most $\frac{\gamma}{400\xi k^2}$. Then by a union bound over at most $\xi k$ groups, the maximum likelihood function assigns $x$ to $\mathcal{G}_i$ with probability at least $1 - \frac{\gamma}{400k}$.

We have that each point is drawn from $G_j$ with probability $\frac{1}{k}$ and so with high probability, at least $\frac{1}{2k}$ fraction of the points will be drawn from $G_j$ and assigned to $\mathcal{G}_i$. $\qquad\square$

**Lemma 3.10.** *Let $C_i$ be the set of points that are drawn from $G_i$ in $\mathcal{D}$. Let $\mathcal{G}_1, \mathcal{G}_2, \ldots$ be the groups of $\widetilde{\mathcal{P}}$. Then with probability $1 - \frac{1}{\mathrm{poly}(k)}$, there exists $\ell \in \mathbb{Z}$ such that for $Q_\ell = \mathcal{G}_\ell \cap C_i$, we have $|Q_\ell| > (1 - \gamma) \max(|\mathcal{G}_\ell|, |C_i|)$.*

*Proof.* By Lemma 3.9, at least $1 - \frac{\gamma}{400k}$ fraction of the points drawn from $G_i$ will be assigned to some $\mathcal{G}_\ell$. By Lemma 3.8, at most $\frac{\gamma}{200\xi k}$ fraction of the point drawn from $G_j$ with $j \neq i$ will be assigned to $\mathcal{G}_\ell$.

Let $n \geq 100k \log(kd)$ be the total number of samples. Then for a uniform mixture, with high probability, the number of points drawn from $G_i$ is at least $\frac{n}{2k}$ and the number of points drawn from other Gaussians is at most $n$, in which case at most $\frac{\gamma n}{200\xi k}$ points from other Gaussians are assigned to $\mathcal{G}_\ell$. Thus with probability $1 - \frac{1}{\mathrm{poly}(k)}$, there exists $\ell \in \mathbb{Z}$ such that for $Q_\ell = \mathcal{G}_\ell \cap C_i$, we have $|Q_\ell| > (1 - \gamma) \max(|\mathcal{G}_\ell|, |C_i|)$. $\qquad\square$

**Lemma 3.11.** *With probability $1 - \frac{1}{\mathrm{poly}(k)}$, Algorithm 3 outputs $k$ groups $\mathcal{G}_1, \ldots, \mathcal{G}_k$.*

*Proof.* Note that each group that does not have at least $\frac{1}{2k}$ fraction of the points is deleted. On the other hand, by Lemma 3.9, we have that with probability $1 - \frac{1}{\mathrm{poly}(k)}$ at most $\frac{\gamma}{400k}$ fraction of the points will not be assigned to the corresponding group. Hence, the total number of these points is at most $\frac{\gamma n}{400k}$. Thus all of the groups that do not correspond to one of the true Gaussians is deleted, and so there are at most $k$ groups, with high probability. $\qquad\square$

**Theorem 1.1.** *Let $\mathcal{D}$ be any $d$-dimensional uniform mixture of Gaussians $G_i, \ldots, G_k$ with $-\log(1 - d_{\mathsf{TV}}(G_i, G_j)) = \Omega(\log(k/\varepsilon))$ for all $i \neq j$. Given a $\beta$-list oracle, there exists an algorithm that takes $n = \mathrm{poly}(dk/\varepsilon)$ samples from $\mathcal{D}$, runs in time $k \cdot \mathrm{poly}(n)$, and returns $k$ hypothesis Gaussians $H_1, \ldots, H_k$ such that with high probability,*

$$d_{\mathsf{TV}}\left(\frac{1}{k} \sum_{i \in [k]} G_i, \frac{1}{k} \sum_{i \in [k]} H_i\right) \leq \tilde{\mathcal{O}}(\varepsilon).$$

*Proof.* Consider Algorithm 3. Let $X$ be the set of $n$ random samples from $\mathcal{D}$. For all $i \in [k]$, let $C_i$ be the set of samples in $S$ that were drawn from $G_i$. Let $\mathcal{G}_1, \mathcal{G}_2, \ldots$ be the groups of $\widetilde{\mathcal{P}}$. Let $\mathcal{E}$ be the event such that:

(1) There are at most $k$ groups $\mathcal{G}_1, \ldots, \mathcal{G}_k$.

(2) For each $i \in [k]$, there exists a corresponding group $\mathcal{G}_\ell$ such that for $Q_\ell = \mathcal{G}_\ell \cap C_i$, we have $|Q_\ell| > (1 - \gamma) \max(|\mathcal{G}_\ell|, |C_i|)$.

Observe that the first property holds With probability $1 - \frac{1}{\mathrm{poly}(k)}$ by Lemma 3.11 and the second property holds with probability $1 - \frac{1}{\mathrm{poly}(k)}$ by Lemma 3.10. Thus by a union bound, we have that $\mathbf{Pr}[\mathcal{E}] \geq 1 - \frac{1}{\mathrm{poly}(k)}$.

Conditioned on $\mathcal{E}$, the sets $\mathcal{G}_1, \ldots, \mathcal{G}_k$ each correspond to $C_1, \ldots, C_k$ up to some permutation $\pi : [k] \to [k]$ and moreover $|\mathcal{G}_{\pi(i)} \cap C_i| > (1 - \gamma) \max(|\mathcal{G}_{\pi(i)}|, |C_i|)$, thus satisfying the conditions of Theorem B.5. Hence by Theorem B.5, the output $\widetilde{\mathcal{D}} = \{H_i\}_{i \in [k]}$ satisfies

$$d_{\mathsf{TV}}\left(\frac{1}{k} \sum_{i \in [k]} G_i, \frac{1}{k} \sum_{i \in [k]} H_i\right) \leq \tilde{\mathcal{O}}(\varepsilon).$$

$\qquad\square$

