# OpenReview forum: "Learning-Augmented Learning of Gaussian Mixture Models"
_ICLR.cc/2025/Conference — Submitted to ICLR 2025_

### Official Review · Reviewer_6UKC · 2024-10-27

**Soundness:** 2
**Presentation:** 2
**Contribution:** 2
**Rating:** 3
**Confidence:** 4

**Summary:**

This paper studies the following problem.
Given samples from a mixture of $k$ $d$-dimensional Gaussians with labels, we would like to return another mixture of $k$ Gaussians such that the total variation of them is at most $\epsilon$.
Here, the label comes with each sample indicating that which component in the mixture the sample is drawn from and the label may not be correct.
Moreover, we assume that the components in the mixture are well-separated.

The result replies on the ROBUSTGAUSSIANS algorithm from (Diakonikolas et al., 2020) which considers the setting of learning mixtures of Gaussians with corrupted samples.
In this paper, the authors exploit the similarity between these two settings and apply ROBUSTGAUSSIANS algorithm to their problem.

The authors show that we need $poly(dk/\epsilon)$ samples to achieve the goal.

**Strengths:**

The paper studies an interesting problem of learning mixtures of Gaussians with extra information. Also, the authors manage to show the polynomial sample complexity.

**Weaknesses:**

It seems that the key component is derived from a previous result (Diakonikolas et al., 2020).
The rest involves using the labels to ensure that the conditions in the previous result hold, allowing us to apply it.
I am not sure if there are fundamentally new techniques introduced.

**Questions:**

.

---

> ### Author Response · Authors · 2024-12-03
>
> > It seems that the key component is derived from a previous result (Diakonikolas et al., 2020). The rest involves using the labels to ensure that the conditions in the previous result hold, allowing us to apply it. I am not sure if there are fundamentally new techniques introduced.
>
> Perhaps there is a misunderstanding -- the only place we use (Diakonikolas et al., 2020) in the main algorithm is to identify the Gaussian underlying each cluster after the partitioning is complete. This is a standard result that can be cited from other references, since we only have a single cluster. In fact, we note in Appendix B that running the result of (Diakonikolas et al., 2020) on the input would result in prohibitive runtime and sample complexity, exponential in $k$.

---

### Official Review · Reviewer_6DYa · 2024-10-27

**Soundness:** 2
**Presentation:** 2
**Contribution:** 2
**Rating:** 6
**Confidence:** 3

**Summary:**

The paper presents an informed strategy for learning a Gaussian mixture model. Given a set of $n$ datapoints sampled from a mixture of $k$ Gaussians, it is assumed that an oracle provides, for each point $i$, a list $P_i$ of $\beta$ possible class labels so that with probability at least $1-\alpha$ the correct label appears in the list. The mixture is then estimated in polynomial time in $n$ up to total variation distance $\tilde{\mathcal O}(\varepsilon)$ if $n=k\\,\text{poly}(d,\ln k,\varepsilon^{-1})$.

**Strengths:**

The algorithm proposed in the paper exemplifies how even possibly corrupted and partial information on the label of data points sampled from a Gaussian Mixture Model can help to construct a Mixture that is close in total variation distance to the true one in polynomial time, as long as the dataset size is large enough. The actual generation of the lists of candidate Gaussians is based on the result in Theorem 3.4, i.e., on a strategy proposed by [Ivkov and Kothari](https://arxiv.org/abs/2206.10942), the main subsequent manipulations being related to the measure of the distances between the generated Gaussians and possible its merging. Nevertheless, the criteria used for merging and the size of the dataset required for the success of the strategy (e.g., having enough points to sensibly sample each class) are under analytical control, which guarantees the statement of the main contribution of the paper.

**Weaknesses:**

A weakness of the paper is the fact that, as the authors state, the strategy very heavily relies on the fact that the objects under analysis are *Gaussian* mixtures. To my understanding, the robustness of the strategy when the underlying mixture is not composed by Gaussian clouds is not obvious. An additional weakness of the paper is, in my opinion, its not outstanding clarity and, at some points, its lack of precision (see the *Questions* section below). This weakness might be fixed after an adequate revision.

**Questions:**

I list below some comments and observations.

- It appears there is a broken sentence in Definition 2.4, "for which $x_i$...?"
- In Definition 3.1 it should be clarified that at least one of the conditions has to hold. Also, the relative-Frobenius closeness relation has to be fixed (there is a very curious typo...)
- In the pseudocode for ```Algorithm 1```, do the authors mean at line 4 "for $i,j\in[|\mathcal L|]$"? The pseudocode is not very clear in expressing what the output is: shouldn't it be the list $\\{\mathcal G_a\\}_{a}$ of grouped pairs? Is the new list $\mathcal L$ made of a single representative for each group (to make then sense of Observation 3.5)?
- In the pseudocode for ```Algorithm 2```, $\mathcal P$ never appears to be used after initialization except to be returned. Also, the authors use the notation $\mathcal P_i$ and $P_i$: please make the text more precise.
- In Lemma 3.8, what is $\gamma$? In Algorithm 3 it seems to be put equal to $\frac{1}{2}$ and never updated. From comparison with the Appendix, this quantity seems to be related to the oracle error rate (in particular, an upper bound to it).
- The ratio behind the choice for $\Delta$ in line 3 of Algorithm 3 stems, from my understanding, from the proof of Lemma 3.8, and similarly the numerical prefactor appearing in Lemma 3.9: could the authors give an intuition behind them? Also, Algorithm 3 refers to $\tilde P_i$ which is, I suppose, the oracle input: is this correct? Moreover, step 13 in Algorithm 3 evokes the ```RobustGaussian``` algorithm in Theorem B.5: I think it would be convenient to mention, at least briefly, this step in the main text and give a proper reference.
- Typo at line 474 (sentence starts with "finds").
- I would like the authors to clarify a little bit more how the generation of the list of Gaussians in the numerical experiment has been performed. From what I understand, the authors neglected the Frobenius separation criterion: is this correct? Also, are numerical experiments performed in $d=2$? Do the authors have a numerical estimate of the decay rate in $n$ of $d_{\rm TV}$ between the ground truth and the reconstructed GMM as $n$ grows?

On a more general question, as the algorithm is fed by a dataset of "oracle-labeled points" I was wondering how the performance of the algorithm would compare with the training error in a "standard" multiclass classification strategy by using, for example, a cross-entropy loss on the dataset by assigning to each datapoint $\boldsymbol x_i$ a label $\boldsymbol y_i=(y_{ij})\_{j\in[k]}$ having components $y_{ij}=\frac{1}{\beta}$ if $i\in \tilde P_j$ and zero otherwise (see, e.g., the high asymptotic analysis in [Loureiro et al. (2021)](https://proceedings.neurips.cc/paper/2021/hash/543e83748234f7cbab21aa0ade66565f-Abstract.html) for the case $n\sim d$). Would this strategy be feasible if one is only interested in the correct labeling of the data points (and not in the reconstruction of the mixture)? How would this approach compare with the proposed one?

As a side note, the paper refers to the additional information exploited by the proposed algorithm to achieve its performance as *learning* provided. To my understanding, the fact that the input predictor is provided by some machine learning strategy plays essentially no role in the results: maybe a more suitable name for the manuscript would have been *Oracle-augmented* more than *Learning-augmented*.

---

> ### Author Response · Authors · 2024-12-03
>
> > In the pseudocode for Algorithm 1, do the authors mean at line 4 "for $i,j\in[|\mathcal{L}|]$"?
>
> No, the code is correct as written because the second for loop iterates over all groups $a$ and $b$
>
> > Is the new list $\mathcal{L}$ made of a single representative for each group (to make then sense of Observation 3.5)?
>
> Yes, that's correct.
>
> > In Lemma 3.8, what is $\gamma$? In Algorithm 3 it seems to be put equal to $\frac{1}{2}$ and never updated
>
> Yes, $\gamma=\frac{1}{2}$
>
> > In the pseudocode for Algorithm 2, $\mathcal{P}$ never appears to be used after initialization except to be returned
>
> $\mathcal{P}$ is the collection $\{\mathcal{P}_i\}$ of partitions $\mathcal{P}_i$.
>
> > The ratio behind the choice for $\Delta$ in line 3 of Algorithm 3 stems, from my understanding, from the proof of Lemma 3.8, and similarly the numerical prefactor appearing in Lemma 3.9: could the authors give an intuition behind them?
>
> Yes, that's correct. The separation needs to be sufficiently high to be detectable.
>
> > Would this strategy be feasible if one is only interested in the correct labeling of the data points (and not in the reconstruction of the mixture)? How would this approach compare with the proposed one?
>
> This is an interesting suggestion, but it seems likely an approach to robustly remove gross misclassifications would still be needed, since a few adversarial points could potentially be problematic.
>
> > I list below some comments and observations.
>
> Thanks for the other comments and observations, they will greatly help the presentation of future versions of the manuscript!

---

### Official Review · Reviewer_iY43 · 2024-11-04

**Soundness:** 2
**Presentation:** 2
**Contribution:** 2
**Rating:** 3
**Confidence:** 4

**Summary:**

In this paper, the authors provides an algorithm for estimation of a GMM up to total variation distance $\epsilon$  that scales linearly with the number of components - this is possible by exploiting auxiliary information which provides a list of labels which has the correct one with probability 1-\alpha. In some sense, the authors claim that this paper breaks the exponential in components lower bound on GMM parameter estimation.

**Strengths:**

The paper is well-written

**Weaknesses:**

The problem is highly fabricated. There is absolutely  no motivation provided to study this problem - in which use case can such a scenario  occur and why should anybody care about a mixture of gaussian study where the number of components is not a constant? It would be great if the authors can provide some motivating examples/ experiments on real world datasets where the number of components is large enough to actually see a difference.

The authors mention that they can use "heuristic such as kmeans++ to cluster the initial data, and use the resulting
centers to form a predictor for the second half of the data” . However such a heuristic does not provide confidence signals to the labels - I am confused regarding this statement with respect to kmeans++. Can the authors explain how kmeans++ or other heuristics could be used to generate the list oracle with confidence signals, as this connection is not immediately clear?

From the example provided in lines 111-123, it seems a robust algorithm for learning mixture of Gaussians will be sufficient - why is this not the case for the general problem?

Theorem 1.1 states guarantees in terms of the Total Variation distance which is different from parameter estimation that has been proved in Moitra and Valiant - is there a reference which says that large TV distance always implies large parameter distance. Why is the result not stated in terms of parameter distance directly is very confusing to me - why is this left to the reader when the authors claim parameter estimation as one of their goals.  Can the authors please restate their main result directly in terms of parameter distance?

In a paper, theorem statements cannot be theorems from other papers (even if cited) - Theorem statements are always main contributions of the paper.  Can the authors clearly distinguish between their novel theoretical contributions and existing results they are building upon - perhaps make different sections including preliminaries?

**Questions:**

Please see above

---

> ### Author Response · Authors · 2024-12-03
>
> > The problem is highly fabricated. There is absolutely no motivation provided to study this problem - in which use case can such a scenario occur and why should anybody care about a mixture of gaussian study where the number of components is not a constant? It would be great if the authors can provide some motivating examples/ experiments on real world datasets where the number of components is large enough to actually see a difference.
>
> There is a large body of work on learning Gaussian mixture models in a semi-supervised setting, such as:
> - Bingchen Zhao, Xin Wen, Kai Han: Learning Semi-supervised Gaussian Mixture Models for Generalized Category Discovery. ICCV 2023: 16577-16587
> - Samet Oymak, Talha Cihad Gulcu: A Theoretical Characterization of Semi-supervised Learning with Self-training for Gaussian Mixture Models. AISTATS 2021: 3601-3609
> - Xiaojin Zhu and Andrew B Goldberg. Introduction to semi-supervised learning. Springer Nature, 2022.
> - Chen Dan, Liu Leqi, Bryon Aragam, Pradeep Ravikumar, Eric P. Xing: The Sample Complexity of Semi-Supervised Learning with Nonparametric Mixture Models. NeurIPS 2018: 9344-9354
> - Xiyu Yu, Tongliang Liu, Mingming Gong, Kayhan Batmanghelich, Dacheng Tao: An Efficient and Provable Approach for Mixture Proportion Estimation Using Linear Independence Assumption. CVPR 2018: 4480-4489
> - Nara M Portela, George DC Cavalcanti, and Tsang Ing Ren. Semi-supervised clustering for mr brain image segmentation. Expert Systems with Applications, 41 (4):1492–1497, 2014
>
> > Can the authors explain how kmeans++ or other heuristics could be used to generate the list oracle with confidence signals, as this connection is not immediately clear?
>
> We use heuristics to create a (likely erroneous) predictor for our algorithms.
>
> > From the example provided in lines 111-123, it seems a robust algorithm for learning mixture of Gaussians will be sufficient - why is this not the case for the general problem?
>
> Non-robust algorithms for learning mixtures of Gaussians already provably require sample complexity and thus exponential in $k$. This work builds upon recent work in the standard setting to achieve robust algorithms for learning mixtures of Gaussians in the semi-supervised setting.
>
> >  is there a reference which says that large TV distance always implies large parameter distance
>
> This is Lemma 3.2 in the manuscript, which is also Proposition A.1 in Bakshi et al. (2020) and Fact 3.24 in Ivkov & Kothari (2022).

---

### Official Review · Reviewer_LefD · 2024-11-13

**Soundness:** 3
**Presentation:** 2
**Contribution:** 3
**Rating:** 5
**Confidence:** 2

**Summary:**

The paper considers the setting of "learning-augmented" algorithms where the sample/computational efficiency of algorithms is improved through the presence of predictions/advice from machine learning models, analogous to the settings of "expert-advice" considered in online learning. The paper proposes algorithms for estimation a Gaussian-mixture model in total-variation distance in the presence of a possibly erroneous list of labels for each sample. The proposed algorithms are polynomial-time and have sample-complex linear in the number of classes and polynomial in dimension $d$. The algorithms rely on seperability assumptions either on the means or the spectrum of the covariances.

**Strengths:**

- The general direction of ``learning-augmented" algorithms is promising.
- The paper is well-written apart from certain consistency issues described in the ``weaknesses" section.
- The paper includes numerical verifications of the proposed algorithm's efficacy.

**Weaknesses:**

- I believe there are certain definitional and consistency issues that need to be addressed. Firstly, from Definition 2.4 it is not clear whether the list of labels correspond to a fixed-known list of means and covariances. From the use of Theorem 3.4 in Algorithm 3 it is clear that this is not the case but this should be transparent in Definition 2.4.
- Second major issue is in Definition 3.1 which defines only when $d_param$ is upper bounded. This definition isn't sufficient to understand what the lower-bound on $d_param$ means in Lemma 3.1. Does it mean that none of the conditions in definition 3.1 are satisfied?
- The proofs are not self-contained and require going through the details of existing works such as Diakonikolas et al. (2017, 2020).

**Questions:**

- How would the results change if the number of classes in not known before-hand?
- In Algorithm 3, why is the list-decoding parameter represented as $\frac{1}{\beta}$ ?
- What is the explicit dependence of the sample-complexity on $\beta,\alpha$?

---

> ### Author Response · Authors · 2024-12-03
>
> > How would the results change if the number of classes in not known before-hand?
>
> A constant-factor upper bound on the number of classes suffices.
>
> > In Algorithm 3, why is the list-decoding parameter represented as $\frac{1}{\beta}$?
>
> Under our assumptions, the number of labels in the list translates to a fraction of uncorrupted samples in the list-decoding setting.
>
> > What is the explicit dependence of the sample-complexity on $\beta,\alpha$?
>
> We assume $\alpha$ and $\beta$ are constants, but their dependencies are not great, with previous references having an exponential dependencies.

---

### Meta-Review · Area_Chair_1WXx · 2024-12-19

**Metareview:**

**Summary of Discussion:**
The reviewers raised concerns about the paper's motivation, novelty, and clarity. The problem setting appears overly contrived, with no clear practical applications or justification for its relevance. The technical contributions rely heavily on prior work, with limited new insights or methodologies introduced. Additionally, the presentation was noted as confusing, with inconsistent definitions and unclear algorithms. While the authors provided theoretical guarantees, these were insufficient to address the reviewers' concerns about the practicality and originality of the approach.

**Conclusion:**
The paper does not meet the standards for acceptance at ICLR due to a lack of clarity, novelty, and compelling motivation.

**Additional Comments On Reviewer Discussion:**

See above.

---

### Decision · Program_Chairs · 2025-01-22

Reject